# Clearance of persistent SARS-CoV-2 associates with increased neutralizing antibodies in advanced HIV disease post-ART initiation

SARS-CoV-2 clearance requires adaptive immunity but the contribution of neutralizing antibodies and T cells in different immune states is unclear. Here we ask which adaptive immune responses associate with clearance of long-term SARS-CoV-2 infection in HIV-mediated immunosuppression after suppressive antiretroviral therapy (ART) initiation. We assembled a cohort of SARS-CoV-2 infected people in South Africa ($n = 994$) including participants with advanced HIV disease characterized by immunosuppression due to T cell depletion. Fifty-four percent of participants with advanced HIV disease had prolonged SARS-CoV-2 infection (>1 month). In the five vaccinated participants with advanced HIV disease tested, SARS-CoV-2 clearance associates with emergence of neutralizing antibodies but not SARS-CoV-2 specific CD8 T cells, while CD4 T cell responses were not determined due to low cell numbers. Further, complete HIV suppression is not required for clearance, although it is necessary for an effective vaccine response. Persistent SARS-CoV-2 infection led to SARS-CoV-2 evolution, including virus with extensive neutralization escape in a Delta variant infected participant. The results provide evidence that neutralizing antibodies are required for SARS-CoV-2 clearance in HIV-mediated immunosuppression recovery, and that suppressive ART is necessary to curtail evolution of co-infecting pathogens to reduce individual health consequences as well as public health risk linked with generation of escape mutants.

While the innate immune response has been shown to be necessary for the initial control of SARS-CoV-2 infection[1,2], adaptive immunity is required for SARS-CoV-2 clearance and is the response targeted by vaccination[3]. Individuals whose adaptive immunity is suppressed have prolonged SARS-CoV-2 infection not observed in immunocompetent people[4–16]. One arm of adaptive immunity is the neutralizing antibody response, which when effective, blocks the virus from infecting cells. Neutralizing antibodies are a correlate of protection against most viral infections[17,18]. Based on vaccine efficacy studies[19–31] against SARS-CoV-

2, neutralizing antibodies correlate strongly with protection from symptomatic infection[18]. Additional evidence that the neutralizing response is critical for protection is that SARS-CoV-2 variants and subvariants evolve mutations which escape neutralizing antibodies made in response to previous infection or vaccination[32].

A second arm of adaptive immunity is the T-cell response. CD4 follicular T-helper cells are required to produce potent neutralizing antibodies through affinity maturation[3]. CD4 T cells also orchestrate the adaptive response to infection and CD8 T cells recognize and kill

✉ e-mail: alex.sigal@ahri.org

infected cells, therefore reducing virus production[33]. However, several factors constrain the selection of viral escape mutations from T-cell-mediated immunity. First, viral antigens are presented according to the HLA repertoire of an individual, which differs within and between populations, and therefore a mutation that leads to T-cell escape in one person may not do so in another[33]. In addition, many of the conserved viral proteins, including accessory and non-structural proteins, provide T-cell epitopes for recognition of infected cells. These factors lead to T-cell responses being generally conserved between SARS-CoV-2 variants, including hypermutated variants such as BA.2.86[34-36]. This may be one reason why disease severity is lower now than at earlier stages in the pandemic[37].

The relative importance of the antibody and T-cell arms of the adaptive immune response is an area of active debate. A related question is whether, when one of the arms is weakened, the other can compensate. B-cell depletion with anti-CD20 therapy leads to persistent SARS-CoV-2 infection[38-41]. In this type of immunosuppression, there is evidence that CD8 T cells compensate for the lack of a B-cell response and clear SARS-CoV-2[42,43].

Case studies of SARS-CoV-2 infections in immunosuppressed individuals show prolonged infection and evolution of genomic changes in the SARS-CoV-2 spike protein associated with escape from neutralizing antibodies[13]. Mutations outside of spike are also common and may confer different functions[44,45]. This is one possible mechanism for how SARS-CoV-2 variants arise[4-16]. Such evolution in immuno-compromised hosts is not unique to SARS-CoV-2 and has been observed in influenza and salmonella infections[46,47].

There can be multiple reasons for immunosuppression[7,16,42,48,49]. One cause of immunosuppression which has been shown in case studies to lead to SARS-CoV-2 long-term infection and evolution is uncontrolled HIV infection resulting in extensive CD4 T-cell depletion, termed advanced HIV disease[4-6,15]. Immunosuppression occurs because HIV infection depletes CD4 T cells by a variety of mechanisms that include death of both HIV-infected[50-53] and uninfected bystander or incompletely infected cells[54-58]. Advanced HIV disease is defined as a CD4 T-cell count lower than 200 cells/μL (a normal CD4 T-cell count is about 1000 cells/μL). This level of CD4 T-cell depletion is known to result in vulnerability to multiple pathogens. One example is *Mycobacterium tuberculosis*, one of the cardinal infections leading to the death of people living with HIV (PLWH) in the pre-antiretroviral therapy (ART) era[59,60]. The number of people globally with immunosuppression because of advanced HIV disease may be considerable. In South Africa alone, the estimated number of PLWH is about 8 million[61]. About 1 in 10 are thought to meet the criteria for advanced HIV disease[62,63]—an estimated 800,000 people. Vaccination could potentially be a strategy to elicit a better immune response to SARS-CoV-2 in people with advanced HIV disease. However, SARS-CoV-2 vaccines were shown to be less effective at eliciting a neutralizing antibody response in PLWH with CD4 counts lower than 200 cells/μL[64-66].

We have previously reported on one case where advanced HIV disease interferes with SARS-CoV-2 clearance and leads to SARS-CoV-2 evolution (participant 27 in this study)[4,5]. The virus which evolved in this participant over 6 months gained immune escape from neutralizing antibodies elicited by SARS-CoV-2 infection and Pfizer BNT162b2 mRNA vaccination. Here we tracked SARS-CoV-2 infection in five participants with advanced HIV disease and failure to adhere to ART, who eventually suppressed HIV viremia. All had prolonged SARS-CoV-2 infection. Among the viruses we isolated from these participants, SARS-CoV-2 originating in a Delta variant infection evolved the most antibody escape mutations and had high-level escape from Delta-elicited neutralizing antibodies. However, it did not escape the current population neutralizing antibody immunity to SARS-CoV-2. Clearance of SARS-CoV-2 was associated with emergence of neutralizing antibodies but not SARS-CoV-2-specific CD8 T-cell responses.

## Results

### Advanced HIV disease leads to long-term SARS-CoV-2 infection and evolution

We assembled an observational longitudinal cohort of SARS-CoV-2 infection in South Africa numbering 994 participants, including 113 PLWH with a CD4 T-cell count lower than 200 cells/μL at enrollment (Supplementary Table S1). During each study visit, a combined naso-pharyngeal and oropharyngeal swab was taken to detect SARS-CoV-2 by qPCR cycle threshold (Ct) where a low Ct indicates high SARS-CoV-2 titer.

We evaluated the length of detectable SARS-CoV-2 infection in advanced HIV disease participants, and a group of non-immunosuppressed participants matched for age and sex to the advanced HIV disease group (Supplementary Table S2). To determine the proportion of participants in each group with prolonged infection and to exclude false-positive results and reinfections, we calculated infection duration for individuals with at least two consecutive SARS-CoV-2 positive qPCR results during the study, followed by at least one sample (positive or negative) a month or more later. The period of infection was taken as the time between the first and last positive qPCR test, where the last positive was followed by two or more negative tests or loss to follow-up. We observed that in the advanced HIV disease group, 54% of infections lasted over 1 month, with some being much longer (Fig. 1A). In contrast, 8% of infections in non-immunosuppressed participants lasted longer than 1 month (Fig. 1A). The difference was significant (Fig. 1A, inset).

We next focused the analysis on five study participants with advanced HIV disease who had long-term SARS-CoV-2 infection and were vaccinated during the study. These participants were between 20 and 42 years of age and had a Covid-19 diagnosis date ranging from September 2020 to December 2021 (Supplementary Table S3; all dates in this and subsequent tables given as month-year). All participants were living with HIV before SARS-CoV-2 infection and participant 255 was HIV-infected by mother-to-child transmission. They were out-patients for 82% of study visits. We included the full inferred course of SARS-CoV-2 infection available to us, including possible reinfections. Therefore, infection periods described below may be longer than those in Fig. 1A for the same participants. The duration of SARS-CoV-2 infection, calculated as the time from first diagnostic to last qPCR positive SARS-CoV-2 test was a median of 207 days, and ranged from 110 to 289 days (Fig. 1B and Supplementary Table S3). The virus was isolated from the swab by outgrowth and/or sequenced when viral titers were sufficient (Ct<30 for sequencing and Ct<25 for isolation). The timepoints and titers of successfully sequenced samples are shown in Fig. 1B as red circles. Four of the participants were enrolled soon after diagnosis. One participant (255) was enrolled in the study in December 2021 during the Omicron infection wave. However, a record of a positive qPCR result for SARS-CoV-2 was present from September 2021, corresponding the Delta variant infection wave (Fig. 1B).

Participants were initiated on ART in line with current national guidelines and received adherence counseling from the clinical team. The ART regimen used was TLD (tenofovir/ lamivudine/ dolutegravir), based on the integrase inhibitor dolutegravir (DTG) combined with the nucleotide/nucleoside reverse transcriptase inhibitors tenofovir (TFV) and lamivudine (3TC). The five participants described here had delayed control of HIV viremia (Fig. 1C). Retrospective detection of ART levels in this group by liquid chromatography coupled with tandem mass spectrometry showed non-adherence to the DTG-based regimen. Green bars in Fig. 1C show the time when DTG started to be consistently detected, and Supplementary Fig. S1 shows study visits where DTG, TFV, and 3TC were detected. Surprisingly, other anti-retroviral drugs were also detected, possibly previously initiated regimens (Supplementary Fig. S1, blue rectangles). The participants eventually adhered to DTG-based ART (Supplementary Fig. S1, green rectangles), leading to HIV suppression observed as a decline in HIV

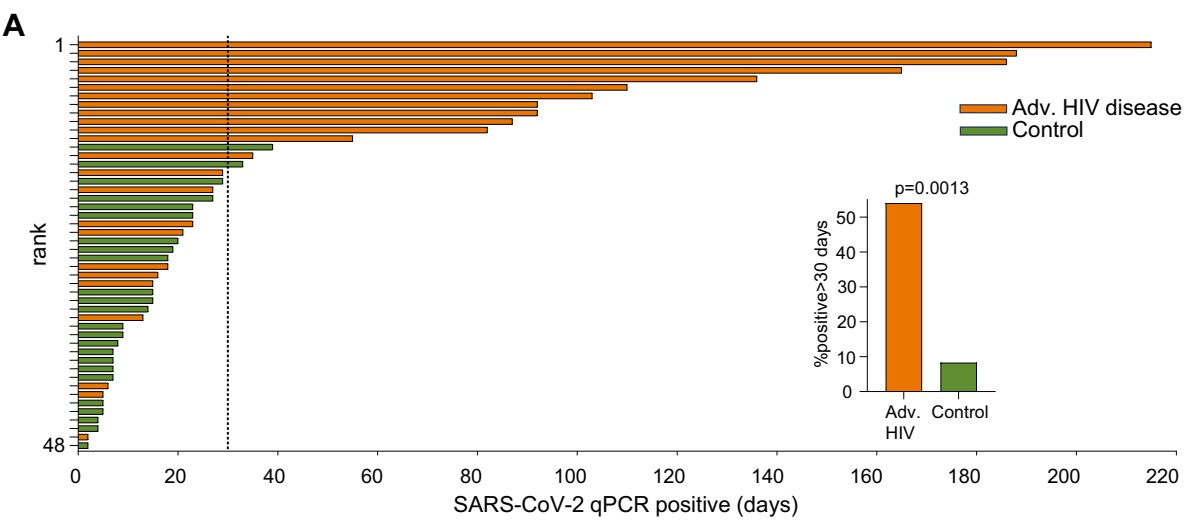

viremia to below the threshold of assay detection (40 HIV RNA copies/ mL) during the study (Fig. 1C).

Four of the participants were vaccinated with two doses of the Pfizer BNT162b2 mRNA vaccine and vaccination times are shown as vertical dashed lines in Fig. 1B, C. The fifth (127) was vaccinated with only one dose because the participant developed synovial inflammation in the wrists and hands 6 days post-first vaccine dose, a rare

adverse event associated with Covid-19 mRNA vaccines[67,68]. The interval between doses approximately followed the South African guidelines at the time of vaccination, which was 6 weeks, although some variation occurred.

We next determined the infecting variant by sequencing. Alignment of sequenced SARS-CoV-2 showed that, consistent with the infection date (Fig. 1B), the participant designated 27 was infected with

**Fig. 1 | Persistent SARS-CoV-2 infection and mutations in advanced HIV disease.**
**A** SARS-CoV-2 duration in 24 advanced HIV disease and 24 non-immunosuppressed participants (controls) matched for age and sex. Infection periods analyzed had at least two consecutive SARS-CoV-2 positive qPCR results, one detecting the full set of assay targets. Possible re-infection periods (positive results separated by two or more negatives) were excluded. $x$ axis is time in days with vertical dashed line denoting 30 days, $y$ axis is rank according to infection duration from longest to shortest. Inset: frequency of SARS-CoV-2 infections lasting for 30 days or more: 11 of 24 (54%) in advanced HIV disease and 2 of 24 (8%) in control participants. $P = 0.0013$ by two-sided Fisher's Exact Test. **B** SARS-CoV-2 infection through time in five participants with advanced HIV disease and vaccination. $x$ axis represents complete infection period, including possible reinfections, and bar above each graph represents the timing of the infection waves for each variant/strain in South Africa. Timeline is continuous and same for all participants shown with ticks on $x$ axis indicating 2 months intervals; the total period covered is the last two months of 2020, all of 2021, and first 8 months of 2022. $y$ axis represents the qPCR cycle threshold (Ct) value, inversely proportional to the SARS-CoV-2 viral titer. Red circles represent successfully sequenced timepoints and vertical dashed lines represent Pfizer BNT162b2 mRNA vaccination times. **C** HIV viral loads for the participants measured in the blood as RNA copies/mL. Green bars above graphs denote periods of adherence to dolutegravir (DTG) based ART. **D** Phylogenetic tree of sequenced virus samples through time for each participant (27: gray circles, 96: purple circles, 127: purple triangles, 127: blue circles, 209: green circles). $x$ axis represents mutation distance from ancestral clade 20A. Potential reinfections in participants 127 and 209 are highlighted with red circles. Tree generated using Nextclade (https://clades.nextstrain.org/).

ancestral SARS-CoV-2 (Fig. 1D, gray circles) and the participant designated 96 was infected with the Beta variant (Fig. 1D, purple circles). Participant 127 was initially infected with a Beta variant (Fig. 1D, purple triangles) but the last sequence was a Delta variant (Fig. 1D, highlighted with red circle), likely a re-infection which occurred during the Delta infection wave in South Africa. Participant 255 was infected with the Delta variant (Fig. 1D, blue circles), consistent with a continuous infection from the first positive diagnostic test in September 2021. Participant 209 was infected with the Omicron BA.1 subvariant (Fig. 1D, green circles), but the sequence from the last timepoint was an Omicron BA.5 subvariant (Fig. 1D, highlighted with red circle), likely a re-infection.

We analyzed non-synonymous changes across the SARS-CoV-2 genome from each sequenced timepoint per participant using the Stanford Coronavirus Antiviral and Resistance Database (https://covdb.stanford.edu/sierra/sars2/by-sequences/). Three of the participants, 27, 96, and 255, had extensive non-synonymous changes in the circulating virus, while two, 127 and 209, had few changes but showed an abrupt shift in sequence consistent with re-infection with a different variant or subvariant (Supplementary Fig. S2). Both 27 and 255 had multiple substitutions in the receptor binding domain (RBD) of the spike gene predicted to lead to escape from neutralizing antibodies. These included E484K, K417T, and F490S[69–73] in the virus from participant 27 and K417T, L452Q, A475V, and E484A[72–76] in the virus from participant 255.

We conclude that these individuals with advanced HIV disease have a significantly prolonged SARS-CoV-2 infection. Second, during the time of prolonged SARS-CoV-2 infection, there is an accumulation of mutations, some of which are known to lead to escape from neutralizing antibodies.

### Clearance of prolonged SARS-CoV-2 infection associates with the emergence of neutralization

The five participants investigated here eventually adhered to antiretroviral therapy and suppressed HIV (Fig. 1C) as well as cleared SARS-CoV-2 (Fig. 1B). HIV suppression results in immune reconstitution and recovery from immunosuppression[77] which may allow for an effective adaptive immune response against SARS-CoV-2. An alternative explanation is that the antiretroviral drugs used to control HIV also inhibit SARS-CoV-2. There have been a number of reports that TFV, a component of the ART regimen given to our study participants and possibly antiretrovirals inhibit SARS-CoV-2 infection[78–80]. Other studies did not observe an association between ART and SARS-CoV-2 clearance[81,82]. To investigate whether the TLD ART regimen used in the study could inhibit SARS-CoV-2, we tested the effect of TFV/3TC/DTG co-formulated ART on SARS-CoV-2 and HIV infection in vitro (Supplementary Fig. S3). We found that this ART regimen potently inhibited HIV infection (Supplementary Fig. S3A, B). However, it had no detectable effect on SARS-CoV-2 (Supplementary Fig. S3B).

We next investigated the relationship between the neutralizing antibody response and SARS-CoV-2 clearance. We isolated and expanded at least one SARS-CoV-2 virus from each participant and tested the neutralizing capacity of the participant's plasma at different timepoints post-diagnosis against the autologous outgrown viruses. Figure 2A shows SARS-CoV-2 viral titers through time for each participant up to and including the final virus clearance timepoint (see Fig. 1B).

The neutralization capacity of participant plasma was determined throughout this work by a focus-forming assay with the live-virus isolates (Supplementary Fig. S4). To quantify the result, we present the focus reduction neutralization test 50 value ($FRNT_{50}$), the inverse of the plasma dilution required for 50% neutralization. Ongoing SARS-CoV-2 infection was correlated with low $FRNT_{50}$ values at or below the level of detection (Fig. 2B). Strikingly, in all the five participants tested, there was a strong increase in neutralization capacity against the autologous viruses at SARS-CoV-2 clearance (Fig. 2B, last timepoint in each graph). The absolute $FRNT_{50}$ value which was associated with viral clearance varied between participants. In participant 96 it also varied between autologous virus isolates. In participant 255, who had undetectable neutralizing antibody levels before SARS-CoV-2 clearance, the $FRNT_{50}$ value became detectable but remained relatively low.

We also tested how different antibody isotypes correlated with SARS-CoV-2 clearance in the five advanced HIV disease participants. Figure S5 compares kinetics of IgG, IgM, and IgA, though IgG was quantified at a dilution of 1:2700, while IgM and IgA, present at much lower levels in plasma, were quantified at a 1:100 dilution. IgG kinetics were most closely associated with clearance, with concentrations in the blood rising at the clearance timepoint or timepoints closely preceding clearance. The kinetics of IgM, indicating the formation of new antibody responses, showed a strong rise in participants 96, 127 and 255, and a slight rise in participant 27 at clearance, but were at much lower concentrations than IgG. IgA kinetics, also much lower than IgG, did not seem to be strongly associated with clearance, with increases close to clearance observed for participants 127 and 255 but not the others.

To examine whether HIV suppression was necessary for clearance, we examined the HIV viral load (Fig. 2C) and CD4 T-cell counts (Fig. 2D). While participants 27, 96, and 127 showed HIV suppression at SARS-CoV-2 clearance, participant 209 did not have full suppression (Fig. 2C, rightmost panel) and participant 255 was unsuppressed, with an HIV viral load of about $10^4$ RNA copies/mL (Fig. 2C, second from right). Immune reconstitution measured in terms of CD4 T-cell counts occurred in every participant (Fig. 2D), although CD4 counts remained at 100 cells/μL or below for four out of five participants. Participants tended to have an increase in CD4 count to the time of SARS-CoV-2 clearance with the exception of participant 255, who showed CD4 T-cell reconstitution which at day 237 post-diagnosis, a time when SARS-CoV-2 was not yet cleared (Fig. 2D, second from right).

In summary, we found an association between SARS-CoV-2 clearance and the emergence of a neutralizing response. Complete HIV suppression was not required, and neutralization was likely mediated by IgG but not IgA isotypes, with the limitation that the

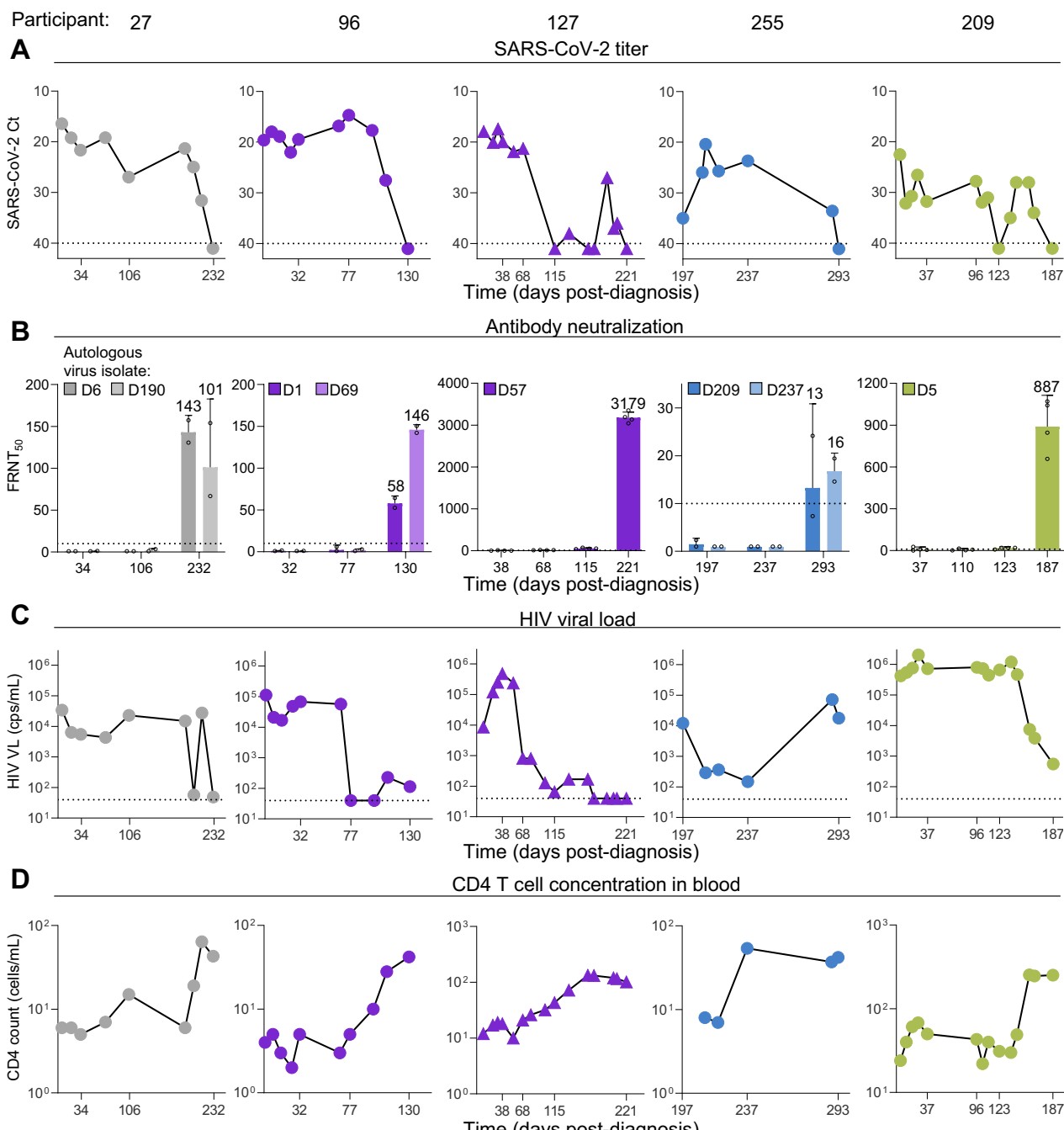

Fig. 2 | **SARS-CoV-2 clearance in advanced HIV disease immunosuppression associates with neutralizing antibody response. A** SARS-CoV-2 titers measured as qPCR Ct through time until SARS-CoV-2 clearance for participants 27, 96, 127, 255, and 209. x axis is time in days post-SARS-CoV-2 diagnosis. y axis is SARS-CoV-2 titer as Ct value. **B** Neutralization of autologous virus by participant plasma sampled at different timepoints. One to two viruses were isolated and tested per participant. Viruses are indicated top left on each graph by day of isolation (denoted by D prefix) post-diagnosis. Four independent experiments were performed per participant. The numbers above bars are geometric mean $FRNT_{50}$, and error bars are geometric mean standard deviations of $FRNT_{50}$ determined from two (two participant viral isolates tested) or four (one isolate tested) independent experiments, with individual experiments shown as points. **C** HIV viral load during SARS-CoV-2 infection period. **D** CD4 T-cell concentrations during SARS-CoV-2 infection period. Horizontal dashed lines represent limits of quantification in all panels.

antibody isotypes were measured in the blood and not at the infection site.

## No association with SARS-CoV-2-specific CD8 T-cell responses

To determine whether SARS-CoV-2-specific T-cell responses were present, we used stimulation with spike peptide pools[36], where the pool used was based on the sequence of the infecting variant. Spike-specific responding T cells were detected using flow cytometry, measuring interferon-gamma (IFN-γ) production (Fig. 3A, see Supplementary Fig. S6A for the gating strategy). First, we tested responses post infection (pre-vaccination) and post-vaccination in five study participants without HIV or with controlled HIV, approximately matched for age and sex to the advanced HIV disease group (Supplementary Table S4). We used a post-vaccination timepoint as we expected vaccination may increase SARS-CoV-2-specific T-cell frequencies. In this group, all participants had SARS-CoV-2-specific CD4

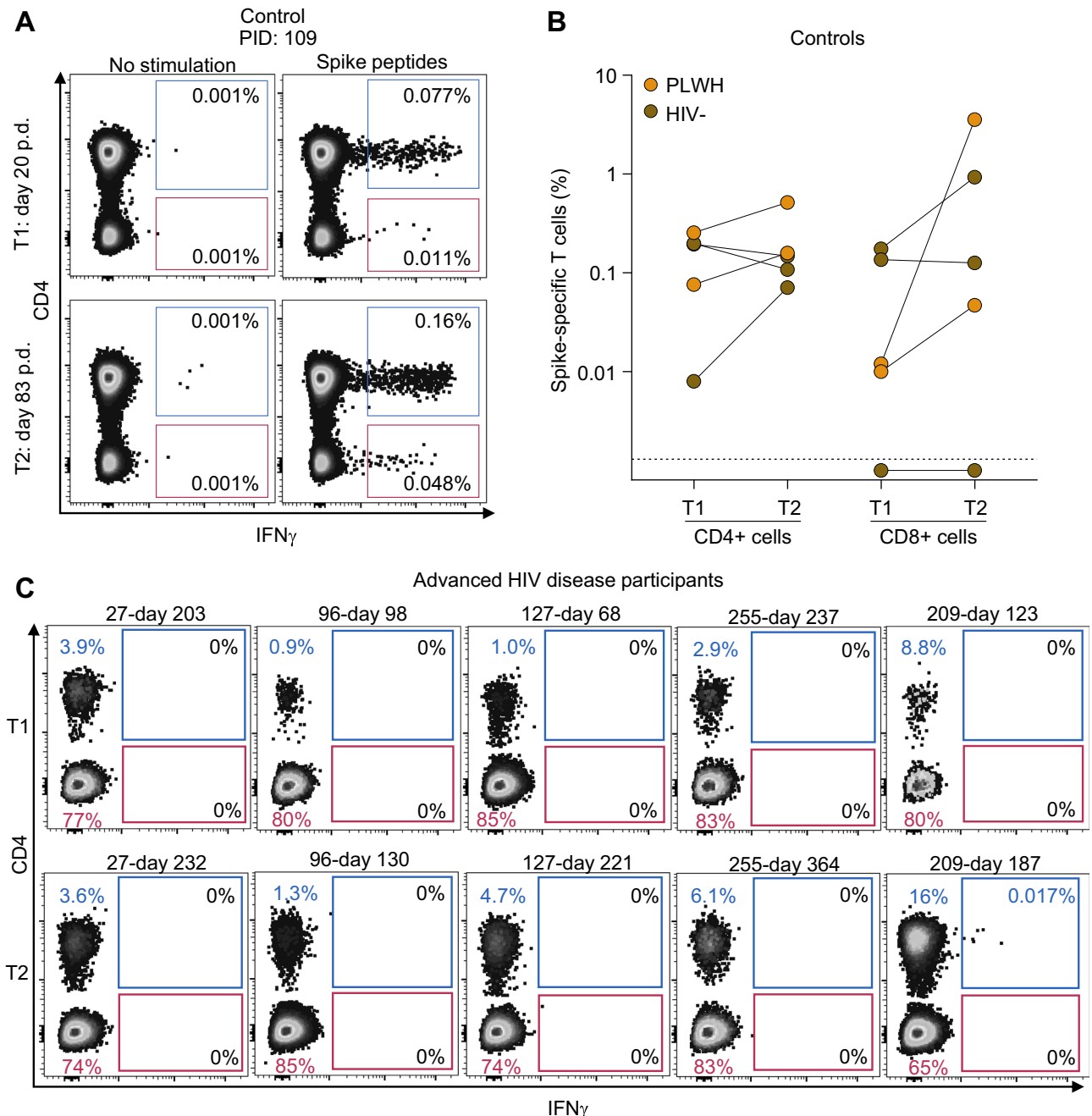

**Fig. 3 | SARS-CoV-2-specific T-cell responses in controls and advanced HIV disease. A** Flow cytometry results showing detection of SARS-CoV-2-specific responses in CD4 and CD8 T cells. T cells were stimulated with spike peptide pools matching the infecting variant and CD4 and CD8 T-cell responses detected by the presence of intracellular interferon-gamma (IFN-γ). The left plot shows unstimulated sample and right plot is 16 h post-peptide stimulation for control participant 109 (PID 109). p.d, days post-diagnosis. **B** CD4 and CD8 T-cell frequencies at two timepoints (T1, post infection; T2, post-vaccination) for each control participant (HIV-uninfected: brown points; controlled HIV: orange points). **C** Flow cytometry results showing CD4 (blue) and CD8 T-cell (red) frequencies in advanced HIV disease participants at two timepoints (T1, pre-SARS-CoV-2 clearance; T2, post-SARS-CoV-2 clearance). There was sufficient PBMC sample for one test per timepoint.

T cells post infection (timepoint designated T1 in Fig. 3A, B), which increased after vaccination for 3 out of 5 participants (timepoint T2 in Fig. 3B; Supplementary Fig. S6 shows flow cytometry results for each participant). CD8 responses were also present post infection in all except one participant (Fig. 3B and Supplementary Fig. S6B).

We next tested for the presence of SARS-CoV-2-specific T cells in the advanced HIV disease participants. Because some of the samples had insufficient cells, we could not choose the same timepoints used to measure neutralizing antibodies. Nevertheless, we were able to test cells from two timepoints, one before and close to SARS-CoV-2 clearance (T1), and one post-clearance, T2 (Supplementary Table S5). As expected, at T1 we observed that the proportion of CD4 T cells was low (ranging from 0.9 to 8.8% of total CD3 + T cells, Fig. 3C). Consistent with ART-associated T-cell reconstitution, at T2 (after ART initiation and in most cases HIV suppression), CD4 frequencies increased in all samples except for participant 27. In contrast, CD8 T cells were preserved at both timepoints, with high cell numbers.

We did not observe SARS-CoV-2-specific CD8 T cells in any of the participants with advanced HIV (Fig. 3C). SARS-CoV-2-specific CD4 T cells were undetectable in all but one sample (participant 209,

T2 sample), where they were present at a low frequency (0.017%). However, we acknowledged that the absence of detectable SARS-CoV-2-specific CD4 T cells in patients with severe lymphopenia may be attributed to the limited number of CD4 + T cells available for flow cytometry analysis. In summary, we did not find evidence that SARS-CoV-2-specific CD8 T-cell responses are involved in SARS-CoV-2 clearance during recovery from advanced HIV-mediated immunosuppression.

## Poor vaccine elicited neutralization in advanced HIV disease participants with HIV viremia

We investigated the neutralizing antibody response elicited by vaccination in the advanced HIV disease participants and compared the response to participants, either PLWH or HIV-negative, who did not have immunosuppression (Supplementary Tables S6 and S7). All participants received the Pfizer BNT162b2 mRNA vaccine.

We tested for neutralizing antibodies against ancestral virus, the Beta, Delta and Omicron BA.1 variants, as well as anti-spike antibody levels, at baseline and after each vaccine dose (Fig. 4A). In the three participants with suppressed HIV at vaccination (27, 96, and 127), there was an increase in neutralization capacity for all strains/variants tested after the first, and if administered, second dose of the vaccine, and a similar increase in overall anti-spike antibodies. In participant 127, who received only one vaccine dose, vaccine elicited neutralization waned quickly, dropping approximately tenfold in about 4 months against all viruses tested. Rapid waning was also seen in participant 27, but only against Omicron BA.1 virus (Fig. 4A).

In participants 255 and 209, HIV viremia was present at vaccination (Fig. 1C). Participant 209 also had SARS-CoV-2 infection at vaccination (second dose, Fig. 1B). Both 255 and 209 showed a seemingly poor neutralization response to the vaccine (Fig. 4A). Neutralization capacity for all strains of participant 255 plasma decreased to below limit of quantification after the first vaccine dose and remained at that level after the second dose (Fig. 4A, second from right panel, with limit of quantification denoted as horizontal dashed line). For 209, neutralization capacity remained below the level of quantification for all viral strains except Omicron BA.1 at two weeks post-second dose, the expected peak of the vaccine response. BA.1 neutralization did increase slightly ($FRNT_{50} = 37$ to $FRNT_{50} = 82$) two weeks post-vaccination (Fig. 4A, right panel). Neutraliztion capacity did increase at later timepoints, either in response to the vaccine, the ongoing SARS-CoV-2 infection, or both.

We compared vaccine responses in five SARS-CoV-2 infected participants without advanced HIV disease. These participants either controlled HIV or were HIV-negative (see Supplementary Table S6 for participant details). This group showed a relatively homogenous response, with a large increase in neutralization after first dose, usually followed by limited waning, then a smaller fold increase in neutralizing capacity post-second dose (Fig. 4B). Anti-spike binding antibody levels mostly mirrored the neutralizing antibody response to the different viral strains in these participants as well as in 4 out of 5 advanced HIV participants. The exception was 255, one of two participants with HIV viremia at vaccination.

We quantified the response post-second dose in a larger group of previously infected participants with no advanced HIV disease (Supplementary Table S7), who were PLWH ($n = 10$) or HIV-negative ($n = 16$). This group included the control group of five participants with detailed longitudinal samples described above. In all participants, we tested neutralization of ancestral virus and the Beta, Delta and Omicron BA.1 variants. All participants without advanced HIV disease showed a marked increase in neutralization of all four strains after vaccination (Fig. 4C). The three advanced HIV disease participants with ART-suppressed HIV at the time of vaccination showed a vaccine-mediated increase in neutralization similar to that of the non-advanced HIV disease participants (Fig. 4C, green lines). In contrast,

neutralization remained low in the two advanced HIV disease participants with unsuppressed HIV (Fig. 4C, red lines).

Therefore, while mRNA vaccination was effective in eliciting a neutralization response in participants without advanced HIV disease as well as those with advanced HIV disease but who already controlled their HIV infection, it did not perform well in eliciting SARS-CoV-2 neutralization in the two advanced HIV disease participants with HIV viremia.

## Hamster infection shows evolved virus from Delta variant infection is antigenically distinct

The degree to which neutralizing antibody immunity elicited by infection with one virus strain can cross-neutralize a second strain is a measure of the antigenic distance between them. However, given that SARS-CoV-2 seroprevalence in South Africa was ~70% pre-Omicron[83], we could not use human sera to measure antigenic distance between recent Omicron subvariants such as XBB and other strains as we would be unlikely to find XBB infected individuals who were uninfected with earlier variants. We have therefore investigated cross-neutralization of ancestral SARS-CoV-2, the Omicron XBB.1.5 subvariant, and two of the evolved SARS-CoV-2 strains with the most antibody escape mutations (Fig. 5A) in the Syrian golden hamster experimental infection model. The evolved viruses tested were the virus isolated after a 190-day infection in participant 27 (27-D190), and the virus isolated from participant 255 after a 237-day infection (255-D237). These were initially ancestral and Delta infections, respectively. Ancestral D614G (B.1 lineage) SARS-CoV-2 and Omicron subvariant XBB.1.5 viruses were also tested. Sixteen days after the experimental infection of hamsters, plasma samples from uninfected and infected animals were assayed against the autologous (infecting) virus as well as the three other viruses to determine neutralization and cross-neutralization (Fig. 5B).

Plasma from uninfected animals did not neutralize any of the viruses (Fig. 5C). Plasma from infected animals most potently neutralized the autologous virus and cross-neutralized the other viruses less well (Fig. 5D–G). The animals not infected with Omicron XBB.1.5 did not have a substantial cross-neutralization of XBB.1.5 (Fig. 5D–F), and animals infected with XBB.1.5 failed to develop a cross-neutralizing response to the other viruses tested (Fig. 5G).

To better visualize the antigenic distances between the viruses, we used antigenic cartography (Fig. 5H) which maps the distances between multiple viruses and the sera elicited by their infections using the Racmacs package[84–86]. Each square of distance on the map corresponds to a twofold drop in neutralizing capacity. Using this visualization, we observed that 27-D190 was antigenically close to the ancestral virus from which it evolved. In contrast, XBB.1.5 was antigenically far from all the other viruses tested. 27-D190, which evolved from ancestral virus, was close to ancestral virus in this hamster infection model. However, the 255-D237 isolate evolved from a Delta infection was antigenically distinct from both ancestral SARS-CoV-2 and Omicron XBB.1.5 (Fig. 5H). Our conclusion from this data is that prolonged SARS-CoV-2 infection in advanced HIV disease immunosuppression can lead to the evolution of SARS-CoV-2 which has extensive antigenic differences relative to both past and currently circulating strains.

## Virus evolved from Delta escapes Delta but not Omicron XBB-elicited neutralization

Currently, many individuals have been infected with multiple SARS-CoV-2 strains/variants. Unlike the singly infected hamsters, they have hybrid immunity. This should broaden the response against new variants[87]. As we have previously characterized the 27-D190 evolved virus[4,45], here we tested the Delta variant evolved 255-D237 virus which had mutations predicted to result in immune escape from neutralizing antibodies (Fig. 5A). We assayed 255-D237 against plasma from participants infected in the Delta, Omicron subvariant BA.1, and Omicron

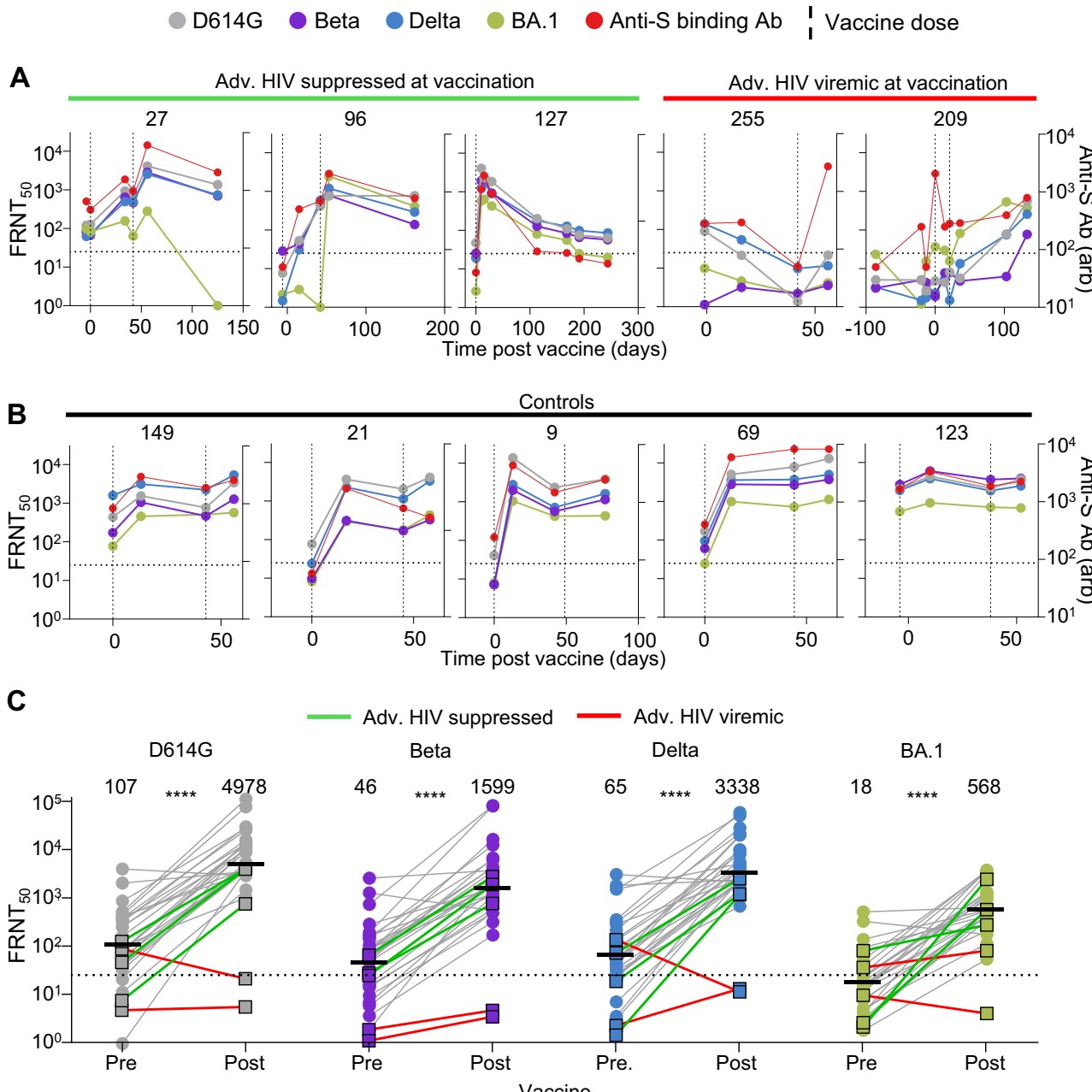

**Fig. 4 | Vaccine responses in advanced HIV disease participants. A** Longitudinal neutralization and anti-spike antibody levels before and after vaccination with the Pfizer BNT162b2 mRNA vaccine in the three advanced HIV disease participants who suppressed HIV at vaccination (27, 96, 127) and the two who did not (255, 209). Neutralization tested against ancestral/D614G SARS-CoV-2 (gray), Beta variant (purple), Delta variant (blue), and Omicron BA.1 subvariant (green). Timing of vaccine doses is represented by vertical dashed lines. *x* axis is time in days post-first vaccine dose and negative numbers represent pre-vaccine period. The left *y* axis is neutralization as $FRNT_{50}$ and right *y* axis is anti-spike antibody level as arbitrary units (arb). **B** Longitudinal neutralization and anti-spike antibody levels before and after vaccination in five participants with no advanced HIV disease. **C** Neutralization of SARS-CoV-2 D614G, Beta, Delta and Omicron BA.1 viral isolates pre-vaccination and after last administered dose by plasma from *n* = 31 participants comprising the two participants with advanced HIV disease and HIV viremia (red lines), the three participants with advanced HIV disease and HIV suppression (green lines) and 26 participants with no advanced HIV disease (gray lines). *y* axis is neutralization as $FRNT_{50}$. The numbers above groups are geometric means and statistical comparison is between participant $FRNT_{50}$ values before and after vaccination. All *P* values were ****$P < 0.0001$ by a two-sided Mann–Whitney test, with exact values being $P = 1 \times 10^{-10}$ (D614G), $2 \times 10^{-9}$ (Beta), $2 \times 10^{-10}$ (Delta), and $9 \times 10^{-12}$ (Omicron BA.1). All $FRNT_{50}$ values are geometric means from two or three independent experiments.

subvariant XBB infection periods in South Africa (see Supplementary Table S8 for participant details).

We first tested 255-D237 against plasma from participants infected with the Delta variant. We observed that, relative to the Delta variant virus, the 255-D237 had over 18-fold lower $FRNT_{50}$ (Fig. 6A), similar in scale to Omicron BA.1 escape relative to

ancestral virus in participants vaccinated with ancestral virus-based vaccines[88]. We then tested this isolate against plasma from Omicron BA.1 infected participants. Consistent with our previous report, we found Omicron BA.1 infection elicited relatively low neutralizing immunity[87]. The 255-D237 evolved virus had a 6.5-fold escape of neutralization relative to the Omicron BA.1 virus, with some of the

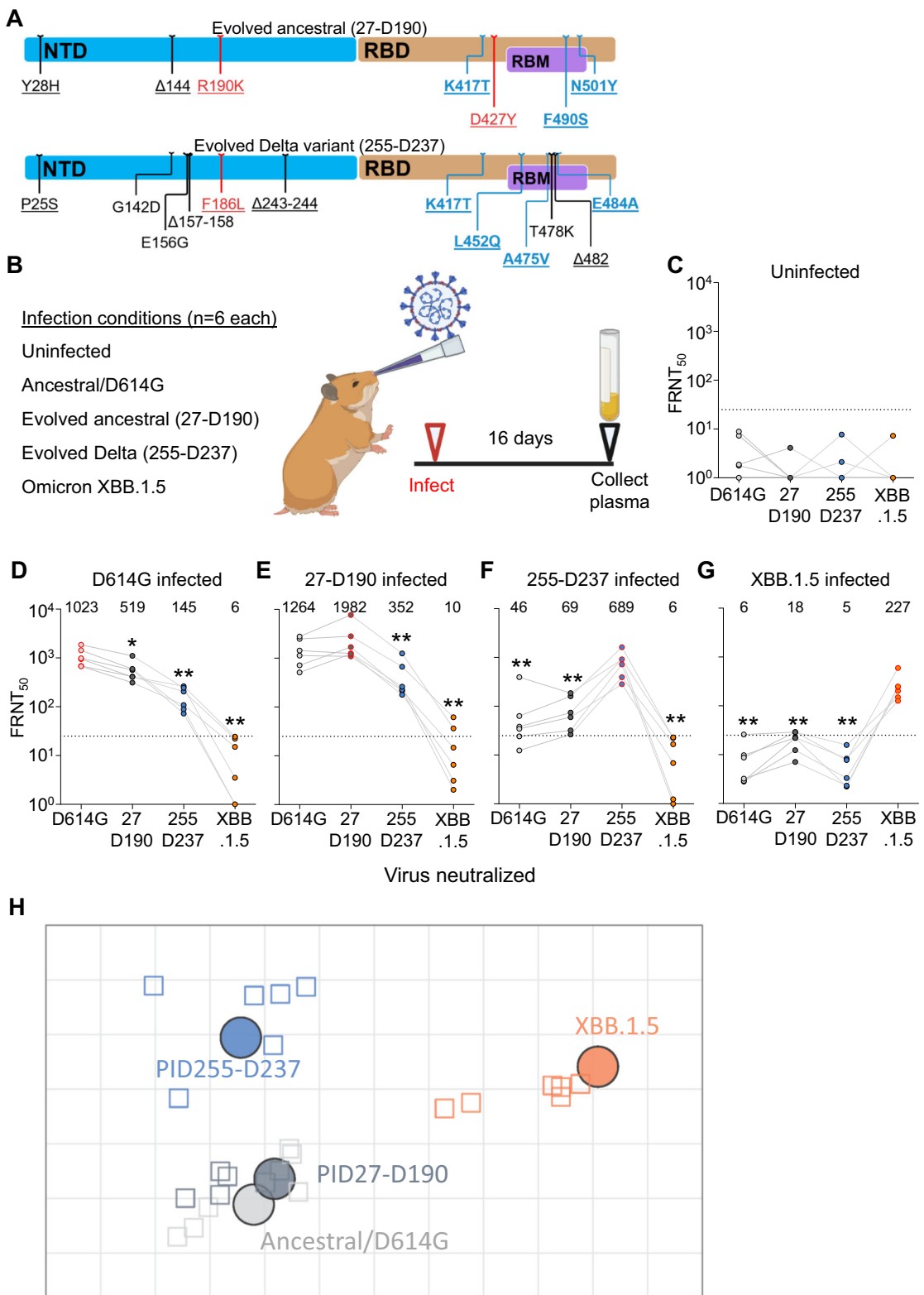

neutralization values for 255-D237 falling below the threshold of quantification (Fig. 6B).

We then tested 255-D237 against plasma from vaccinated participants with breakthrough Omicron BA.1 infection. As we previously reported[87], the vaccinated/BA.1 infected group had stronger Omicron BA.1 virus neutralization as well as better cross-neutralization of other variants compared to the BA.1 infected unvaccinated group. We observed that vaccination coupled with BA.1 infection-elicited

neutralization was able to neutralize the 255-D237 virus to a similar extent as the BA.1 virus (Fig. 6C). Lastly, we assayed 255-D237 against plasma from individuals infected during the period when Omicron XBB subvariants were dominant in South Africa (from November-December 2022, https://www.nicd.ac.za/diseases-a-z-index/disease-index-covid-19/sars-cov-2-genomic-surveillance-update/ accessed August 8, 2023). These participants are also expected to have immunity from previous infections, including pre-Omicron variants. In this

**Fig. 5 | Antigenic distances of SARS-CoV-2 evolved from ancestral and Delta infections in the hamster model. A** Substitutions and deletions in the N-terminal domain (NTD) and receptor binding domain (RBD) of SARS-CoV-2 spike in 27-D190 and 255-D237. Blue mutations: known antibody escape. Red mutations: mutations with global prevalence below 0.01%. RBM: receptor binding motif. Representation and characterization based on the Stanford Coronavirus Antiviral and Resistance Database (https://covdb.stanford.edu). **B** Schematic of hamster infection experiment. Six animals in two independent experiments were used per infection condition with $FRNT_{50}$ values determined once for each animal. Parts of (**B**) created with BioRender.com. **C** Neutralization of ancestral D614G, 27-D190, 255-D237, and Omicron XBB.1.5 subvariant viruses in uninfected hamsters. **D−G** Neutralization of

the same viruses at 16 days post infection in hamsters infected with ancestral/ D614G (**D**), 27-D190 (**E**), 255-D237 (**F**), and the XBB.1.5 subvariant (**G**). Numbers above the points and horizontal bars are geometric means for the group. Significance was determined by a two-sided Mann−Whitney test relative to the autologous (infecting) virus. Significant $P$ values from left to right were: 0.03, 0.002, 0.002. 0.009, 0.002. 0.009, 0.002, 0.002, 0.002, 0.002, 0.002. **H** Antigenic map of neutralization data presented in (**D−G**). Virus strains/variants are shown as colored circles and hamster plasma samples as open squares with the color corresponding to the infecting virus. Each square on the grid corresponds to a twofold decrease in neutralization. Map created using Racmacs (https://acorg.github.io/Racmacs/).

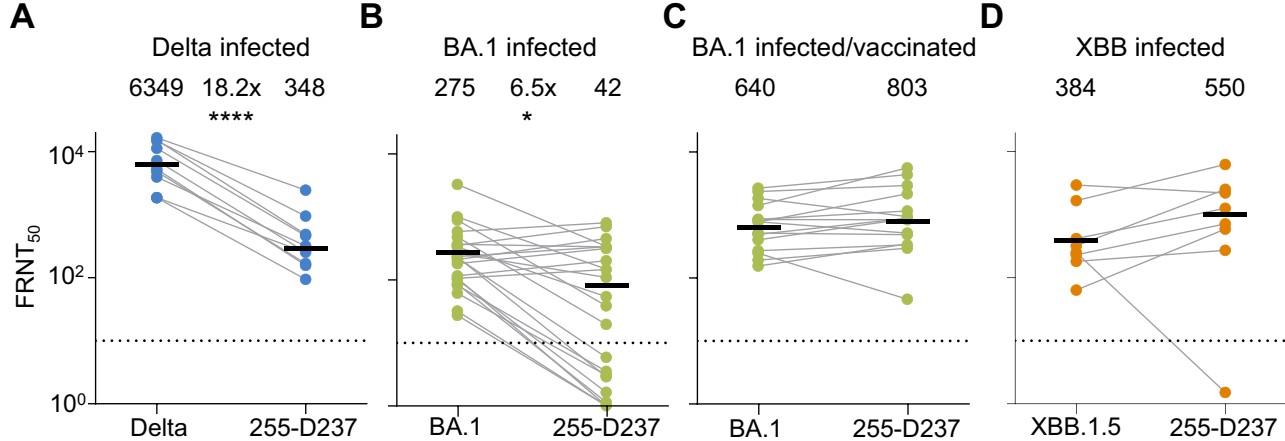

**Fig. 6 | Neutralization of Delta evolved virus by participants with Delta and Omicron infection elicited immunity. A** Neutralization of 255-D237 virus compared to Delta variant virus by plasma samples from ten participants infected during the Delta infection wave in South Africa. ****$P = 4 \times 10^{-5}$ by a two-sided Mann−Whitney test. **B** Neutralization of the 255-D237 compared to Omicron BA.1 subvariant virus in plasma samples from 24 participants infected with Omicron BA.1. $FRNT_{50}$ for one participant was out of the range of the graph but included in the calculation of the geometric mean. *$P = 0.03$ by a two-sided Mann−Whitney test.

**C** Neutralization of 255-D237 compared to Omicron BA.1 virus in plasma samples from 15 vaccinated participants with Omicron BA.1 breakthrough infection. **D** Neutralization of the 255-D237 compared to the Omicron XBB.1.5 subvariant virus in plasma from eight participants infected during XBB derivative infection period in South Africa. All samples collected approximately 2−3 weeks post-diagnosis. The numbers above the points and horizontal bars are geometric means for the group with $FRNT_{50}$ values determined once for each participant/virus combination.

group, there was no escape of the 255-D237 virus relative to the XBB.1.5 subvariant (Fig. 6D).

In summary, our results show that while the most evolved virus sampled in this study was antigenically distinct from the Omicron XBB.1.5 subvariant, this virus did not escape neutralization resulting from XBB-derived subvariant infections or relatively recent hybrid immunity.

## Discussion

Here we characterized five prolonged SARS-CoV-2 infections in individuals with advanced HIV disease and HIV viremia at study enrollment. Infections lasting over a month were common in this group but rare in participants without advanced HIV disease. SARS-CoV-2 clearance was associated with the emergence of a neutralizing antibody response which did not need to be very strong, as demonstrated by participant 255. The emergence of the neutralizing antibody response was associated with CD4 T-cell reconstitution, as measured by CD4 T-cell concentrations in the blood, in four out of five advanced HIV disease participants. However, in participant 255, CD4 T-cell reconstitution peaked at day 237 post-SARS-CoV-2 diagnosis, whereas the neutralizing response and virus clearance happened on day 293. At least for this case, CD4 T-cell reconstitution was not sufficient for virus clearance, or clearance was delayed relative to CD4 reconstitution. HIV viremia did not need to be completely suppressed, although T-cell reconstitution likely occurred because of ART initiation.

In contrast to neutralizing antibody responses, no SARS-CoV-2-specific CD8 responses were detected, and there were too few CD4 T cells to determine whether a SARS-CoV-2-specific CD4 T-cell response was present. This was not the case in control participants without immunosuppression who were SARS-CoV-2 infected and later vaccinated. SARS-CoV-2-specific CD8 T cells were readily detected in three out of four of these participants. Since T-cell help is essential to generate an effective antibody response[89], CD4 T-cell reconstitution may have led to the production of effective neutralizing antibodies, although the frequency of these CD4 T cells was too low to detect in the peripheral blood. Based on these results, it seems that in the case of advanced HIV disease immunosuppression and unlike with therapeutic anti-CD20 depletion of B cells[42,43], CD8 T-cell responses do not compensate for the lack of neutralizing antibody responses. Moreover, SARS-CoV-2-specific CD8 T-cell function seems slower to recover from HIV-mediated immunosuppression compared to the B-cell-mediated neutralizing antibody response.

We did not observe any effect of the antiretroviral drug regimen on SARS-CoV-2 infection and there was little association between clearance and IgA levels. A consequence of the prolonged infection was the evolution of extensive neutralization escape from the infecting variant. Reassuringly, this did not lead to escape from neutralizing immunity elicited by the more current Omicron XBB-derived subvariant infections.

We also found that the Pfizer BNT162b2 mRNA vaccine was effective in increasing binding and neutralizing antibodies against SARS-CoV-2 variants in participants with advanced HIV disease who suppressed HIV viremia. Based on the lack of substantially elicited neutralization, vaccination was not effective in the two advanced HIV disease participants with HIV viremia. This agrees with previous studies showing less effective neutralizing antibody responses to vaccines in individuals with low CD4 T-cell counts[64–66]. These data therefore support the principle of starting ARVs in parallel with or prior to SARS-CoV-2 vaccination in patients with HIV.

In conclusion, we have found evidence that neutralizing antibodies associate with SARS-CoV-2 clearance and are likely required for such clearance to happen in recovery from HIV-mediated immunosuppression. Furthermore, while SARS-CoV-2 can evolve extensive neutralization escape in prolonged infection in advanced HIV disease, current population immunity provides a substantial barrier against viruses evolved in this way. Study limitations include that the number of participants with advanced HIV disease and persistent HIV viremia was small.

It is likely that long-term infection and evolution can occur with other pathogens in individuals with advanced HIV disease. Therefore, investment in an effective global HIV treatment strategy may be necessary to reduce the chances that this type of evolutionary process occurs in a pathogen with pandemic potential.

## Methods

### Informed consent and ethical statement
This was an observational study with longitudinal sample collection. Vaccination against SARS-CoV-2 and antiretroviral treatment for HIV infection were part of clinically indicated care. No interventions were administered as part of the study. All blood samples used for neutralization studies, nasopharyngeal swabs from the advanced HIV disease participants for outgrowth and sequencing, as well as nasopharyngeal swabs for isolation of the ancestral/D614G, Beta, and Delta virus were obtained after written informed consent from adults with PCR-confirmed SARS-CoV-2 infection enrolled in a prospective cohort of SARS-CoV-2 infected individuals at the Africa Health Research Institute approved by the Biomedical Research Ethics Committee at the University of KwaZulu-Natal (reference BREC/00001275/2020). Participants were reimbursed for each visit, based on time, inconvenience and expenses as approved in the protocol. The Omicron/BA.1 virus was isolated from residual swab used for diagnostic testing (Witwatersrand Human Research Ethics Committee reference M210752). The nasopharyngeal swab for isolation of the Omicron XBB.1.5 subvariant was collected after written informed consent as part of the COVID-19 transmission and natural history in KwaZulu-Natal, South Africa: Epidemiological Investigation to Guide Prevention and Clinical Care in the Centre for the AIDS Programme of Research in South Africa (CAPRISA) study and approved by the Biomedical Research Ethics Committee at the University of KwaZulu-Natal (reference BREC/00001195/2020, BREC/00003106/2021).

### Clinical laboratory testing
SARS-CoV-2 Ct and HIV viral load quantification was performed from a nasopharyngeal swab universal transport medium aliquot and 4-ml EDTA tube of blood, respectively, at an accredited diagnostic laboratory (Molecular Diagnostic Services, Durban, South Africa). The CD4 count was performed by an accredited diagnostic laboratory (Ampath, Durban, South Africa).

### Detection of ART concentrations in plasma by LC-MS/MS
Sample analysis was performed using an Agilent High-Pressure Liquid Chromatography (HPLC) system coupled to the AB Sciex 5500, triple quadrupole mass spectrometer equipped with an electrospray ionization (ESI) TurboIonSpray source. The LC-MS/MS method was developed and optimized for the quantitation of tenofovir, lamivudine, and dolutegravir in the same sample. A protein precipitation extraction method using acetonitrile was used to process 50 µL plasma samples. In total, 50 µL of water and 50 µL of ISTD solution was added, and the sample was briefly mixed. In total, 150 µL of acetonitrile was subsequently added to facilitate protein precipitation, vortex mixed, and centrifuged at $16,000 \times g$ for 10 min at 4 °C. Overall, 170 µL of the clear supernatant was then transferred to a clean micro-centrifuge tube and dried down using a SpeedVac dryer set at 40 °C. The dried samples were then reconstituted in 100 µL of 0.02% sodium deoxycholate (Sigma) in Millipore filtered water, vortex mixed, briefly centrifuged, placed in a small insert vial, capped, placed in the autosampler compartment (maintained at 4 °C) and analyzed using LC-MS/MS. The analytes were separated on an Agilent Zorbax Eclipse Plus C18 HPLC column using gradient elution. The column oven was set at 40 °C, a sample volume of 2 µL was injected and the flow rate was set to 0.2 mL/min. Mobile phase A consisted of water with 0.1% formic acid, and B consisted of acetonitrile with 0.1% formic acid. The drug analytes were monitored using multiple-reaction monitoring mode for positive ions except for efavirenz which was monitored in the negative ion scan mode. Analyst software, version 1.6.2 was used for quantitative data analysis. Blanked values for EFV, FTC and TFV were in the range of 3 ng/mL, and this was set as the detection limit.

### Whole-genome sequencing
RNA was extracted on an automated Chemagic 360 instrument, using the CMG-1049 kit (Perkin Elmer, Hamburg, Germany). The RNA was stored at −80 °C prior to use. Libraries for whole-genome sequencing were prepared using either the Oxford Nanopore Midnight protocol with Rapid Barcoding or the Illumina COVIDseq Assay. For the Illumina COVIDseq assay, the libraries were prepared according to the manufacturer's protocol. Briefly, amplicons were tagmented, followed by indexing using the Nextera UD Indexes Set A. Sequencing libraries were pooled, normalized to 4 nM and denatured with 0.2 N sodium acetate. An 8 pM sample library was spiked with 1% PhiX (PhiX Control v3 adaptor-ligated library used as a control). We sequenced libraries on a 500-cycle v2 MiSeq Reagent Kit on the Illumina MiSeq instrument (Illumina). On the Illumina NextSeq 550 instrument, sequencing was performed using the Illumina COVIDSeq protocol (Illumina Inc, USA), an amplicon-based next-generation sequencing approach. The first strand synthesis was carried using random hexamers primers from Illumina and the synthesized cDNA underwent two separate multiplex PCR reactions. The pooled PCR-amplified products were processed for tagmentation and adapter ligation using IDT for Illumina Nextera UD Indexes. Further enrichment and cleanup was performed as per protocols provided by the manufacturer (Illumina Inc). Pooled samples were quantified using Qubit 3.0 or 4.0 fluorometer (Invitrogen Inc.) using the Qubit dsDNA High Sensitivity assay according to the manufacturer's instructions. The fragment sizes were analyzed using TapeStation 4200 (Invitrogen). The pooled libraries were further normalized to 4 nM concentration and 25 µL of each normalized pool containing unique index adapter sets were combined in a new tube. The final library pool was denatured and neutralized with 0.2 N sodium hydroxide and 200 mM Tris-HCL (pH7), respectively. In all, 1.5 pM sample library was spiked with 2% PhiX. Libraries were loaded onto a 300-cycle NextSeq 500/550 HighOutput Kit v2 and run on the Illumina NextSeq 550 instrument (Illumina, San Diego, CA, USA). For Oxford Nanopore sequencing, the Midnight primer kit was used as described by Freed and Silander55. cDNA synthesis was performed on the extracted RNA using LunaScript RT mastermix (New England BioLabs) followed by gene-specific multiplex PCR using the Midnight Primer pools which produce 1200 bp amplicons which overlap to cover the 30-kb SARS-CoV-2 genome. Amplicons from each pool were pooled and used neat for barcoding with the Oxford Nanopore Rapid Barcoding kit as per the manufacturer's protocol. Barcoded samples were pooled and bead-purified. After the bead cleanup, the library was

loaded on a prepared R9.4.1 flow-cell. A GridION X5 or MinION sequencing run was initiated using MinKNOW software with the base-call setting switched off. We assembled paired-end and nanopore.fastq reads using Genome Detective 1.132 (https://www.genomedetective.com) which was updated for the accurate assembly and variant calling of tiled primer amplicon Illumina or Oxford Nanopore reads, and the Coronavirus Typing Tool. For Illumina assembly, GATK HaploTypeCaller --min-pruning 0 argument was added to increase mutation calling sensitivity near sequencing gaps. For Nanopore, low coverage regions with poor alignment quality (<85% variant homogeneity) near sequencing/amplicon ends were masked to be robust against primer drop-out experienced in the Spike gene, and the sensitivity for detecting short inserts using a region-local global alignment of reads, was increased. In addition, we also used the wf_artic (ARTIC SARS-CoV-2) pipeline as built using the nextflow workflow framework. In some instances, mutations were confirmed visually with.bam files using Geneious software V2020.1.2 (Biomatters). The reference genome used throughout the assembly process was NC_045512.2 (numbering equivalent to MN908947.3). To determine which SARS-CoV-2 proteins were mutated, sequence was input into the sequence analysis application in the Stanford Coronavirus Antiviral and Resistance Database (https://covdb.stanford.edu/sierra/sars2/by-sequences/) with HTML as output. Mutations were then visualized in Excel relative to the infecting variant.

## Phylogenetic analysis

Sequences were aligned by Nextclade version 2.9.1 (https://clades.nextstrain.org/). The json file output from the Nextclade analysis was loaded into Auspice (https://auspice.us/). Visualization was filtered to include reference sequences from clades 20A, 20B, 20H (Beta), 21J (Delta), 21M, 21K, 21L, and 22B (Omicron), and the input sequences (new nodes), for a combined 1408 genomes. The tree was then filtered to show new nodes only. Tip labels were removed and SVG downloaded for final processing using Microsoft Powerpoint software.

## Cells

The H1299-E3 (H1299-ACE2, clone E3) cell line used in the live-virus infections was derived from H1299 (CRL-5803) as described in previous work[88,90] and propagated in growth medium consisting of complete Roswell Park Memorial Institute (RPMI, Gibco, 21875-034) with 10% fetal bovine serum (Hyclone, SV30160.03) containing 10 mM of hydroxyethylpiperazine ethanesulfonic acid (HEPES, Lonza, 17-737E), 1 mM sodium pyruvate (Gibco, 11360-039), 2 mM L-glutamine (Lonza BE17-605E) and 0.1 mM nonessential amino acids (Lonza 13-114E). HEK293 cells (CRL-1572) used to produce HIV were grown in Dulbecco's Modified Eagle's Medium (DMEM, Gibco 41965-039) with 10% fetal bovine serum containing 10 mM of HEPES, 1 mM sodium pyruvate (Gibco, 11360-039), 2mM L-glutamine (Lonza BE17-605E) and 0.1 mM nonessential amino acids. The RevCEM-GFP HIV infection reporter cell line was obtained from the AIDS Research and Reference Reagent Program, National Institute of Allergy and Infectious Diseases, National Institutes of Health from Y. Wu and J. Marsh. The cell line was subcloned (see description in ref. 91) to increase the maximum fraction of cells with GFP fluorescence upon HIV infection and clone B8 used for the assays. RevCEM-GFP cells were grown in RPMI 1640 with 10% fetal bovine serum containing 10 mM of HEPES, 1 mM sodium pyruvate, 2 mM L-glutamine, and 0.1 mM nonessential amino acids.

## HIV antiretroviral therapy resuspension and incubation

One Acriptega tablet (Mylan Pharmaceuticals) containing 50 mg dolutegravir (DTG), and 300 mg each of lamivudine (3TC) and tenofovir disoproxil fumarate (TDF), the components of the first line TLD regimen. The pill was powdered using a pill crusher and powder resuspended in 25 mL of N,N-Dimethylformamide (DMF, Sigma D4551) and Dulbecco's phosphate-buffered saline (DPBS, Whitehead Scientific

PBS-1A) at a dilution of 1:4 (DMF:PBS) for a stock concentration of 2 mg/mL DTG and 12 mg/mL 3TC/TDF. Solution was maintained at 37 °C for 4 h until powder completely dissolved. TLD stocks were serially diluted in RPMI in a 2-fold dilution series and the diluted TLD was added to H1299-E3 cells plated in a 96-well plate (Corning CLS, 3595) at 20,000 cells per well for SARS-CoV-2 infection or to RevCEM-GFP cells in a 24-well plate (TPP, 92024) at $0.5 \times 10^6$ cells per well for HIV infection. Both cell types were incubated 1 day with TLD before infection with the respective virus.

## HIV-1 production and infection of the RevCEM-GFP reporter cell line

The HIV molecular clone pNL4-3 was obtained from the AIDS Research and Reference Reagent Program, National Institute of Allergy and Infectious Diseases, National Institutes of Health from M. Martin. HIV NL4-3 stock was produced by transfection of HEK293 cells with pNL4-3 using the TransIT-LT1 (Mirus,) transfection reagent. Virus-containing supernatant was harvested after 2 days of incubation and filtered through a 0.45 μm filter. For infection, RevCEM-GFP reporter cells were plated in a 24-well plate at $0.5 \times 10^6$ cells per well. NL4-3 virus was added at a dilution of 1:20 from stock to each well, except for uninfected controls. Forty-eight hours post infection, cells were collected, centrifugated and resuspended in Cytofix/Cytoperm (BD Biosciences, 554655) for 20 min at 4 °C in the dark. Cells were washed twice in BD wash buffer and analyzed on an FACSymphony flow cytometer (BD). Data were analyzed using FlowJo and Graphpad Prism 10.9.0 software.

## Live-virus neutralization assay and testing of the TLD ART regimen on SARS-CoV-2 infection

H1299-E3 cells were plated in a 96-well plate (Corning) at 30,000 cells per well 1 day pre-infection. Plasma was separated from EDTA-anticoagulated blood by centrifugation at 500×g for 10 min and stored at −80 °C. Aliquots of plasma samples were heat-inactivated at 56 °C for 30 min and clarified by centrifugation at 10,000×g for 5 min. Virus stocks were used at ~50–100 focus-forming units per microwell and added to diluted plasma. Antibody–virus mixtures were incubated for 1 h at 37 °C, 5% CO₂. Cells were infected with 100 μL of the virus–antibody mixtures for 1 h, then 100 μL of a 1× RPMI 1640 (Sigma-Aldrich, R6504), 1.5% carboxymethylcellulose (Sigma-Aldrich, C4888) overlay was added without removing the inoculum. Cells were fixed 18 h post infection using 4% PFA (Sigma-Aldrich, P6148) for 20 min. Foci were stained with a rabbit anti-spike monoclonal antibody (BS-R2B12, GenScript A02058) at 0.5 μg/mL in a permeabilization buffer containing 0.1% saponin (Sigma-Aldrich, S7900), 0.1% bovine serum albumin (BSA, Biowest, P6154) and 0.05% Tween-20 (Sigma-Aldrich, P9416) in PBS. Plates were incubated with primary antibody overnight at 4 °C, then washed with wash buffer containing 0.05% Tween-20 in PBS. A secondary goat anti-rabbit HRP conjugated antibody (Abcam ab205718) was added at 1 μg/mL and incubated for 2 h at room temperature with shaking. TrueBlue peroxidase substrate (SeraCare 5510-0030) was then added at 50 μL per well and incubated for 20 min at room temperature. Plates were imaged in an ImmunoSpot Ultra-V S6-02-6140 Analyzer ELISPOT instrument with BioSpot Professional built-in image analysis (C.T.L). For testing of the HIV TLD regimen on the SAR-CoV-2, 20,000 cells were used per well, and cells were incubated with TLD 1 day pre-infection. At infection, SARS-CoV-2 resuspended in the TLD concentration tested was added to the cells. Downstream steps were as for the live-virus neutralization assay.

## Statistics and fitting

All statistics and fitting were performed using custom code in MATLAB v.2019b. Neutralization data were fit to:

$$Tx = 1/1 + (D/ID_{50}). \quad (1)$$

Here Tx is the number of foci at plasma dilution D normalized to the number of foci in the absence of plasma on the same plate. $ID_{50}$ is the plasma dilution giving 50% neutralization. $FRNT_{50} = 1/ID_{50}$. Values of $FRNT_{50} < 1$ are set to 1 (undiluted), the lowest measurable value. We note that the most concentrated plasma dilution was 1:25 and therefore $FRNT_{50} < 25$ were extrapolated.

### Luminex-based isotyping of antibody responses

Serum samples were heat-inactivated for 60 min at 56 °C. Samples were diluted in 1% milk; 5% goat sera; 0,05% Tween-20 in 1×PBS. Samples were diluted in four threefold dilutions from a starting dilution of 1:100. For the detection of total IgG and IgM an in-house control of pooled plasma was used, diluted, and titrated in the same way as the samples. Antibody responses were measured against full-length SARS-COV-2 spike variants (D614G, Beta, Delta, and Omicron BA.1 in 1×PBS). Magnetic beads were diluted in BAMA wash buffer (1% BSA, 0,05% Tween-20; 0.05% sodium azide in 1×PBS). In total, 50 µL of the bead mixture was added per well per plate. The beads and test samples were incubated at room temperature at 300 rpm for 120 min. Plates were washed three times with 250 µL BAMA wash buffer. For the detection of the different isotypes, 50 µL of 0.65 µg phycoerythrin (PE)-conjugated secondary detection antibodies were added (Mouse-Anti-Human IgM-PE (Southern Biotech, cat no 9020-09), Goat Anti-Human IgA-RPE (Bio-rad, Ref 205009) and total IgG (Goat Anti-Human IgG-fc (Invitrogen, 12-4998-82)) and incubated for 60 min. The plates were washed three times on a Biotek plate washer. Thereafter bead mixtures were resuspended in 100 µL BAMA wash buffer and read on a Bio-Plex 200 system using the Bio-plex manager.

### SARS-CoV-2 spike enzyme-linked immunosorbent assay (ELISA)

Two µg/mL of spike protein was used to coat 96-well, high-binding plates and incubated overnight at 4 °C. The plates were incubated in a blocking buffer consisting of 5% skimmed milk powder, 0.05% Tween-20, 1× PBS. Plasma samples were diluted to 1:100 starting dilution in a blocking buffer and added to the plates. The secondary antibody was diluted to 1:3000 or 1:1000, respectively, in blocking buffer and added to the plates followed by TMB substrate (Thermofisher Scientific). Upon stopping the reaction with 1 M $H_2SO_4$, absorbance was measured at a 450 nm wavelength. MAbs CR3022 and BD23 were used as positive controls and Palivizumab was used as a negative control.

### Cell stimulation and flow cytometry staining for T-cell assays

Cryopreserved peripheral blood mononuclear cells (PBMC) were thawed, washed, and rested for 4 h in RPMI 1640 (Sigma-Aldrich) supplemented with 10% heat-inactivated FBS. After resting, cells were seeded in a 96-well V-bottom plate at -0.5 to $1 × 10^6$ cells/well. Cells were stimulated with custom-made SARS-CoV-2 mega pools spanning the entire Spike protein of the ancestral, Beta, Delta or Omicron variants (1 µg/mL), provided by Dr Alessandro Sette (La Jolla Institute for Immunology, USA). All stimulations were performed in the presence of Brefeldin A (10 µg/mL, Sigma-Aldrich) and co-stimulatory antibodies against CD28 (clone 28.2) and CD49 (clone L25) (1 µg/mL each; BD Biosciences). As a negative control, PBMC were incubated with co-stimulatory antibodies, Brefeldin A, and an equimolar amount of DMSO. After 16 h of stimulation, cells were washed, stained with LIVE/DEAD™ Fixable Near-IR Stain (Invitrogen) and subsequently fixed and permeabilized using Cytofix/Cytoperm buffer (BD Biosciences). Cells were then stained with CD3 BV785 (OKT3, Biolegend), CD4 PE-Cy7 (L200, BD Biosciences), CD8 BV510 (RPA-8, Biolegend) and IFN-γ Alexa 700 (B27, BD Biosciences). After staining, cells were washed and fixed in 1% paraformaldehyde (ThermoFisher Scientific). Samples were acquired on a BD Fortessa flow cytometer using FACSDiva software and analyzed using FlowJo (v10, FlowJo LLC). The gating strategy is presented in Figure S6. All data are presented after background subtraction.

### SARS-CoV-2 hamster infections

Golden Syrian hamsters (*Mesocricetus auratus*), 4–5 weeks old, were purchased from Charles River Laboratories, USA. Experimental work was approved by the Animal Ethics Committee at the University of KwaZulu-Natal (reference: REC/00004197/2022). $n = 30$ hamsters (19 female, 11 male, 15 animals per experiment with two experiments performed) were used. Six animals were used per infection condition. Infections were carried out in the animal biosafety level 3 containment facility. Hamsters were lightly sedated with 3% isoflurane (Piramal Healthcare, Mumbai, India) and infected with virus by intranasal inoculation of 50 µL per nostril of virus solution. Plasma of infected animals and uninfected controls was collected at 16 days post infection using cardiac puncture under anesthesia with 5% isoflurane. Hamsters were immediately euthanized post-puncture with 1 mL of 200 mg/mL sodium pentobarbitone solution (Bayer AG, Leverkusen, Germany). Plasma was separated by centrifugation at 1000×*g* for 10 min. Aliquots of plasma samples were heat-inactivated at 56 °C for 30 min and clarified by centrifugation at 10,000×*g* for 5 min.

### Reporting summary

Further information on research design is available in the Nature Portfolio Reporting Summary linked to this article.

## Data availability

Viral isolates are available upon reasonable request. Sequences of isolated SARS-CoV-2 used in this study have been deposited in GISAID and GenBank with accession numbers as follows: D614G (B.1 lineage), EPI_ISL_602626.1 (GISAID), OP090658 (GenBank). BA.1 (B.1.1.529.1), EPI_ISL_7886688, OP090659. BA.4, EPI_ISL_12268495.2, OP093374. BA.5, EPI_ISL_12268493.2, OP093373. XBB.1.5, EPI_ISL_17506815, OR782922. BA.2.86, EPI_ISL_18226980, OR775659. Beta (B.1.351), EPI_ISL_678615, OR936719. Delta (B.1.617.2), EPI_ISL_3118687, OR936720. 0027-D6, EPI_ISL_15541746, OR936722. 0027-D20, EPI_ISL_15541747, OR936723. 0027-D34, EPI_ISL_15541748, OR936751. 0027-D71, EPI_ISL_15541749, OR936770. 0027-D106, EPI_ISL_15541750, OR936771. 0027-D190, EPI_ISL_2397313, OR936772. 0096-D1, EPI_ISL_14666761, OR936774. 0096-D15, EPI_ISL_14666763. 0096-D32, EPI_ISL_13986492, OR936848. 0096-D68, EPI_ISL_18030390, OR939248. 0096-D77, EPI_ISL_18030391, OR939249. 0096-D110, EPI_ISL_14666766. 0127-D10, EPI_ISL_16508746, OR939365. 0127-D24, EPI_ISL_16508747, OR939364. 0127-D31, EPI_ISL_16508748, OR939446. 0127-D38, EPI_ISL_16508749, OR939591. 0127-D54, EPI_ISL_18030392, OR939646. 0127-D68, EPI_ISL_16508751, OR939648. 0127-D192, EPI_ISL_14666773, OR939692. 0255-D209, EPI_ISL_14599778, OR939724. 0255-D211, EPI_ISL_14599779, OR939726. 0255-D219, EPI_ISL_14599780, OR939727. 0255-D237, EPI_ISL_13986497, OR939737. 0209-D5, EPI_ISL_18030393, OR939740. 0209-D26, EPI_ISL_18030394, OR939739. 0209-D144, EPI_ISL_12970433, OR939741. 0209-D159, EPI_ISL_14666777, OR939743. Source data are provided with this paper.

## Code availability

Script in MATLAB v.2019b to fit neutralization data for $FRNT_{50}$ is available on GitHub (https://github.com/sigallab/NatureMarch2021).

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

## Acknowledgements

This study was supported by the Bill and Melinda Gates Award INV-018944 (A.S.), Wellcome Trust award 226137/Z/22/Z (A.S., W.A.B., and P.L.M.), the South African Medical Research Council (A.S. and P.L.M.), National Institute of Health award 5R01AI138546-05, and the United World Antivirus Research Network through National Institute of Health subaward UWSC14272 BPO70432 (A.S.). P.L.M. is supported by the South African Research Chairs Initiative of the Department of Science and Innovation and National Research Foundation of South Africa. The funders had no role in study design, data collection and analysis, decision to publish, or preparation of the manuscript. A.S. wishes to thank Ron Milo, Karen Makar, and Thandi Onami for discussion and suggestions on analysis and presentation of results, Nono Mkhize for input on Luminex analyses, and Alessandro Sette and Alba Grifoni for providing the Spike SARS-CoV-2 peptide pools.

## Author contributions

A.S. and F.K. conceived the study and designed the study and experiments. F.K., K.K., M.B., J.U., G.L., Q.A.K., S.S.A.K., M.H., and A.v.G. identified and provided virus samples. C.R., R.K., and W.A.B. performed and analyzed the T-cell experiments. S.v.G., Z.M., and P.L.M. performed antibody binding and isotyping experiments. Y.S.M., F.K., B.I.G., M.B., K.K., No.M., Ni.M., M.M., and N.N. set up and managed the cohort and cohort data. Y.G. and K.T. performed animal experiments. K.K., Z.J., K.R., Y.G., E.V., Z.M., and K.G. performed experiments and sequence analysis with input from A.S., A.v.G., T.d.O., and R.J.L. A.S., F.K., M.B., J.U., R.J.L., and G.L. interpreted data with input from Y.S.M., S.S.A.K., W.H., and T.d.O. A.S., F.K., and G.L. prepared the manuscript with input from all authors.

## Competing interests

The authors declare no competing interests.

## Additional information

Farina Karim[1,2], Catherine Riou [3,4], Mallory Bernstein [1], Zesuliwe Jule[1], Gila Lustig[5], Strauss van Graan[6,7], Roanne S. Keeton[3], Janine-Lee Upton[1], Yashica Ganga[1], Khadija Khan [1,2], Kajal Reedoy[1], Matilda Mazibuko[1], Katya Govender [1], Kershnee Thambu[1], Nokuthula Ngcobo[1], Elizabeth Venter[6,7], Zanele Makhado[6,7], Willem Hanekom [1,8], Anne von Gottberg [9,10], Monjurul Hoque[11], Quarraisha Abdool Karim [5,12], Salim S. Abdool Karim [5,12], Nithendra Manickchund[13], Nombulelo Magula[14], Bernadett I. Gosnell [13], Richard J. Lessells [5,15], Penny L. Moore [6,7], Wendy A. Burgers [3,4], Tulio de Oliveira [5,15,16,17], Mahomed-Yunus S. Moosa [13] & Alex Sigal[1,2,5] ✉

[1]Africa Health Research Institute, Durban, South Africa. [2]School of Laboratory Medicine and Medical Sciences, University of KwaZulu-Natal, Durban, South Africa. [3]Institute of Infectious Disease and Molecular Medicine, Division of Medical Virology, Department of Pathology, University of Cape Town, Observatory, South Africa. [4]Wellcome Centre for Infectious Diseases Research in Africa, University of Cape Town, Observatory, South Africa. [5]Centre for the AIDS Programme of Research in South Africa, Durban, South Africa. [6]SAMRC Antibody Immunity Research Unit, School of Pathology, Faculty of Health Sciences, University of the Witwatersrand, Johannesburg, South Africa. [7]National Institute for Communicable Diseases of the National Health Laboratory Service, Johannesburg, South Africa. [8]Division of Infection and Immunity, University College London, London, UK. [9]Centre for Respiratory Diseases and Meningitis, National Institute for Communicable Diseases, a division of the National Health Laboratory Service, Johannesburg, South Africa. [10]School of Pathology, University of the Witwatersrand, Johannesburg, South Africa. [11]KwaDabeka Community Health Centre, KwaDabeka, South Africa. [12]Department of Epidemiology, Mailman School of Public Health, Columbia University, New York, NY, USA. [13]Department of Infectious Diseases, Nelson R. Mandela School of Clinical Medicine, University of KwaZulu-Natal, Durban, South Africa. [14]Department of Internal Medicine, Nelson R. Mandela School of Medicine, University of Kwa-Zulu Natal, Durban, South Africa. [15]KwaZulu-Natal Research Innovation and Sequencing Platform, Durban, South Africa. [16]Centre for Epidemic Response and Innovation, School of Data Science and Computational Thinking, Stellenbosch University, Stellenbosch, South Africa. [17]Department of Global Health, University of Washington, Seattle, WA, USA. ✉e-mail: alex.sigal@ahri.org

