## [Peer Review File · Nature Communications]

Clearance of persistent SARS-CoV-2 associates with increased neutralizing antibodies in advanced HIV disease post-ART initiationREVIEWER COMMENTS

Reviewer #1 (Remarks to the Author):

This is an important paper outlining virus evolution and serological responses in HIV-infected individuals with prolonged SARS-CoV-2 infection. The evolution of new variants has been hypothesized to be most likely to be the result of long-term infection in immune suppressed individuals for some time, based on case report data and case series. This has been shown both in B and to a lesser extent in T-cell deficiency. The literature on the impact of HIV and T-cell deficiency has been lacking and there have been no systematic studies previously. Importantly, the impact of ARV therapy on immune restoration during chronic SARS-CoV-2 infection has not been well documented.

Notably, the Beta and Omicron variants most likely emerged in Southern Africa and were detected in South Africa due to impressive sequencing efforts in this area. This part of the African region is home to a large number of individuals infected with HIV, a significant number of whom have significant CD4+ cell depletion and subsequent immune deficiency.

This paper describes 5 patients (from a cohort of 994 patients, 113 of whom were heavily immune suppressed) with HIV-related immune deficiency and prolonged SARS-CoV-2 detection, 2 of whom had reinfection and 3 of whom had one variant detected.

The notable findings, carried out using robust methodology are of clearance on appearance of neutralizing antibodies and binding antibodies that also correlated with detection of dolutegravir in patient samples. Also, the emergence of virus evolution and immune escape in chronic infection (mostly in previously well-described sites) is also a very important observation. The use of in vitro assays of patient samples and also hamsters to look at the impact of XBB 1.5 antibodies is very useful, alongside the inclusion of appropriate controls.

This study highlights the risk associated with untreated or poorly treated HIV infection in the generation of variants with a shift in antigenic profile and resultant changes in phenotype. It also highlights the impact of ARV (dolutegravir-based) therapy in reversing this risk.

The manuscript and figures are highly polished and clear. I have a few fairly minor questions/comments.

1. How many people in the HIV+ COVID cohort became chronically infected with SARS-CoV-2 - are we in a position to comment on this risk in more detail? The systematic design of this study is really important and the associated risk has not previously been quantified, while case studies/series have previously been reported (although none quite as well as this). I note also from the protocol that transmission was considered. Were the highly evolved lineages transmitted?
2. Minor point. In the x axis of Figure 1 timelines. are each of the intervals correctly labelled or have they slipped a bit? The 2021 timeline seems to start with a short period and then there another 4 - are these meant to be continuous timelines e.g. ~2 months each??
3. Given that HIV causes T cell deficiency, it would have been useful to see some data on ELISpot or flow cytometry responses as well as the neutralization data - this should be acknowledged as a limitation of the study.

Reviewer #2 (Remarks to the Author):

This manuscript by Karim et al. describes the long-term SARS-CoV-2 infection in people living with HIV. HIV-mediated immunosuppression can alter the SARS-CoV-2 infection dynamics, and virus clearance can be delayed significantly. They have shown that in people with advanced HIV disease, SARS-CoV-2 infection can persist for 100-200 days. Interestingly, they found that suppressing HIV viremia using ART led to the clearance of SARS-CoV-2. This clearance was associated with the emergence of neutralizing antibodies. They also report that the SARS-CoV-2 vaccine was effective only when HIV has been suppressed. People with active viral replication or low CD4+ T cell counts failed to respond to the vaccine. This study highlights how immunosuppressed people, particularly those living with advanced HIV disease, can be persistently infected with SARS-CoV-2. Authors have shown that neutralizing antibodies are associated with protection against SARS-CoV-2, as shown by several other studies. Importantly, the level of HIV suppression and CD4+ count in HIV-infected people might offer a way to predict vaccine efficacy in these individuals. Overall, the authors used a unique cohort of patients and have done a longitudinal study to understand SARS-CoV-2 infection dynamics. The findings reported here will be helpful to further understand the interplay between HIV and SARS-CoV-2 infection. As long-term COVID is now becoming a public health problem, this study will offer some insight to further understand SARS-CoV-2 persistence and clearance in immunosuppressed individuals.

I believe the manuscript can be further improved, and there are some key issues that need to be addressed.

Major points:

1) The clearance of SARS-CoV-2 is associated with an increase in FRNT titer, a decrease in HIV genome copies, and an increase in CD4 count. The authors have highlighted that clearance was associated with an increase in neutralizing activity. However, it might be possible that CD4+ cell increase may lead to increased CD8+ activity. Is it possible to look at CD8+ T cell activity in these individuals?

2) It is not clear why neutralizing antibodies appear after 100-200 days of infection. One explanation is that an increase in CD4+ T cells might have improved B cell affinity maturation, leading to potent neutralizing antibody production. However, those individuals might still be responding to infection and generating non-neutralizing antibodies even when they have low levels of CD4+ T cells. It will be important to look at different antibody isotypes (IgM, IgG, and IgA) to see if those individuals were making antibodies to SARS-CoV-2.

3) The clearance of SARS-CoV-2 and HIV RNA coincides with the initiation of ART. Is there any direct effect of ART on SARS-CoV-2? Has it been shown that ART does not inhibit SARS-CoV-2 replication? If not, an in vitro test can be performed to show the effect of these drugs on SARS-CoV-2.

Minor points

1) Line 50 – the sentence seems incomplete

2) Line 66-67 – the sentence needs to be rephrased

3) Show correlation plots to show the correlation between SARS-CoV-2 genome copies, neutralizing antibodies, CD4+ T cell count, and HIV genome copies.

4) Fig 2. X-axis label missing

5) How many times FRNT were performed?

6) Can you include the conclusion for every result section? The way it is written now makes it very hard to follow the paper. If you include a conclusion for every section, it would be easier for the readers to transition from one section to another.

Reviewer #3 (Remarks to the Author):

The study follows 5 participants who live with advanced HIV disease and are immunosuppressed over their courses of SARS-COV-2 infection. Data are presented for their HIV infection, viremia control, CD4 cells counts, and their SARS-COV-2 infection and clearance, vaccination, and binding and neutralizing antibody responses. The study provide strong evidence the association between neutralizing antibody response development and SARS-COV-2 virus clearance in these participants, and also demonstrated the need for HIV viremia control and sufficient CD4 T cell presence for the development of function neutralizing response against SARS-COV-2.

Data are clearly presented. Conclusions drawn are well-supported. Findings of substantial evolution of SARS-COV-2 virus in immunosuppressed subjects over long infection period is consistent with other researchers finding. The strong association between neutralizing response and SARS-COV-2 virus clearance is important for the field. The observations that a "normal" vaccine responses can be elicited in people living with advanced HIV disease provided that viremia is controlled and CD4 T cell population is reconstituted should be informational for clinical practices involving advance HIV infections.

Questions:

1. The association between timing of neutralizing antibody response and virus clearance if clear in the study. However, timing of neutralizing antibody response also coincide with CD4 T cell reconstitution in most cases. Have SARS-COV-2 specific T cell responses being studied in these participants? HIV viremia control can be associated with both better CD4 health and CD8 T cell response. It would be great if we can see whether T cell response is also involved in the SARS-COV-2 clearance. I do understand such investigation can be limited by specimen availability though.

2. Neutralizing antibody response development is associated with CD4 T cell reconstitution in most cases. Therefore CD4 T cell count also strongly associated with virus clearance in most cases. However in Pt 255, CD4 T cells returned to normal range at D237, whereas neutralizing response and virus clearance didn't happen until D293. It may worth noting in discussion that at least in this case, CD4 T cell reconstitution itself isn't sufficient for virus clearance.

Minor comments:

3. Line 193 "Participant 255 cleared SARS-CoV-2 at the vaccine baseline visit..." and then Line 195 "Participant 255 still had SARS-CoV-2 infection at vaccination". This is very confusing. Is the Participant 255 on line 195 supposed to be Participant 209?

4. Line 198 "but not at two weeks post-vaccination, the first post-vaccination timepoint tested and expected peak....". Is this referring to only after the first vaccination? It looks like neutralizing response did increase at the first time point tested after the 2nd vaccination.

5. Line 211 "and this was also observed for participant 209". This refers to "strong increase in anti-spike antibodies but not neutralization". However for Pt 209, there is not really a strong increase in anti-spike antibodies following vaccination. The peak binding response that shows an increase is at Day 0 of 1st vaccination, which is most likely from infection. The binding response did not increase until much later on.

15 December 2023

We thank the Reviewers for the insightful comments. We have now revised the manuscript to address them.

This includes:

- 1) Measurement of the SARS-CoV-2 specific T cell response in advanced HIV disease
- 2) Analysis of the frequency of long-term SARS-CoV-2 infection in participants with advanced HIV disease
- 3) Characterization of antibody isotypes
- 4) In vitro investigation of the effect of antiretroviral drugs on SARS-CoV-2 infection

We feel these revisions have greatly improved the work. Below are point-by-point responses to the Reviewer comments.

Reviewer #1

This is an important paper outlining virus evolution and serological responses in HIV-infected individuals with prolonged SARS-CoV-2 infection.

We thank the Reviewer for the support.

The evolution of new variants has been hypothesized to be most likely to be the result of long-term infection in immune suppressed individuals for some time, based on case report data and case series. This has been shown both in B and to a lesser extent in T-cell deficiency. The literature on the impact of HIV and T-cell deficiency has been lacking and there have been no systematic studies previously. Importantly, the impact of ARV therapy on immune restoration during chronic SARS-CoV-2 infection has not been well documented.

We thank the Reviewer for supporting our approach.

The notable findings, carried out using robust methodology are of clearance on appearance of neutralizing antibodies and binding antibodies that also correlated with detection of dolutegravir in patient samples. Also, the emergence of virus evolution and immune escape in chronic infection (mostly in previously well-described sites) is also a very important observation. The use of in vitro assays of patient samples and also hamsters to look at the impact of XBB 1.5 antibodies is very useful, alongside the inclusion of appropriate controls.

We thank the Reviewer for the support.

This study highlights the risk associated with untreated or poorly treated HIV infection in the generation of variants with a shift in antigenic profile and resultant changes in phenotype. It also highlights the impact of ARV (dolutegravir-based) therapy in reversing this risk.

We thank the Reviewer for supporting our approach.

DURBAN

K-RITH Tower Building, 719 Umbilo Road, Durban
Private Bag X7, Congella, 4013, South Africa
T +27 (0)31 260 4991
E durban@ahri.org

SOMKHELE

Africa Centre Building, via R618 to Hlabisa, Somkhele, Mtubatuba
PO Box 198, Mtubatuba, 3935, South Africa
T +27 (0)35 5507500
E somkhele@ahri.org

www.ahri.org  
Africa Health Research Institute (AHRI)
is the operational name of K-RITH (NPC),
a Registered Non-Profit Company 2011/011985/08

The manuscript and figures are highly polished and clear. I have a few fairly minor questions/comments.

We thank the Reviewer for the support.

1. How many people in the HIV+ COVID cohort became chronically infected with SARS-CoV-2 - are we in a position to comment on this risk in more detail?

This is an important point and we have now analyzed infection in all participants with advanced HIV disease in our cohort (n=24) as well as control participants matched for age and sex (n=24). To avoid false positives or reinfections we set strict criteria: Infection periods analyzed had at least two consecutive SARS-CoV-2 positive qPCR results, one detecting the full set of assay targets. Possible re-infection periods (positive results separated by two or more negatives) were excluded. Also, to be included in the analysis, these positive samples needed to be followed by at least one sample (positive or negative) a month or more later. The results are presented in a new Figure 1A:

These results show that over half of the advanced HIV disease participants have SARS-CoV-2 infection for a month or more (usually much longer), but this is rare in the immunocompetent participants. We describe this in the results, lines 121-132:

“We evaluated the length of detectable SARS-CoV-2 infection in advanced HIV disease participants and a group of non-immunosuppressed participants matched for age and sex to the advanced HIV disease group (Table S2). To determine the proportion of participants in each group with prolonged infection and to exclude false-positive results and re-infections, we calculated infection duration for individuals with at least two consecutive SARS-CoV-2 positive qPCR results during the study, followed by at least one sample (positive or negative) a month or more later. The period of infection was taken as the time between the first and last positive qPCR test, where the last positive was followed by two or more negative tests or loss to follow-up. We observed that in the advanced HIV disease group, 54% of infections lasted over one month with some being much longer (Figure 1A). In contrast, 8% of infections in non-immunosuppressed participants lasted longer than 1 month (Figure 1A). The difference was significant (Figure 1A inset).”

The systematic design of this study is really important and the associated risk has not previously been quantified, while case studies/series have previously been reported (although none quite as well as this). I note also from the protocol that transmission was considered. Were the highly evolved lineages transmitted?

We agree transmission would be very informative but unfortunately the study design cannot answer this – we did not do contact tracing because our resources were limited.

2. Minor point. In the x axis of Figure 1 timelines. are each of the intervals correctly labelled or have they slipped a bit? The 2021 timeline seems to start with a short period and then there another 4 - are these meant to be continuous timelines e.g. ~2 months each??

We thank the Reviewer for pointing this out. We have now clarified the timelines in the Figure 1 legend:

“Timeline is continuous and same for all participants shown with ticks on x-axis indicating two months intervals; the total period covered is the last two months of 2020, all of 2021, and first 8 months of 2022.”

3. Given that HIV causes T cell deficiency, it would have been useful to see some data on ELISpot or flow cytometry responses as well as the neutralization data - this should be acknowledged as a limitation of the study.

We have now done this, results presented in a new Figure 3. The conclusion is that we do not detect SARS-CoV-2 specific CD8 T cell responses after clearance, indicating that they may not be associated with clearance. We could not determine whether there are SARS-CoV-2 specific CD4 responses because the concentration of these cells was too low:

Reviewer #2:

Overall, the authors used a unique cohort of patients and have done a longitudinal study to understand SARS-CoV-2 infection dynamics. The findings reported here will be helpful to further understand the interplay between HIV and SARS-CoV-2 infection. As long-term COVID is now becoming a public health problem, this study will offer some insight to further understand SARS-CoV-2 persistence and clearance in immunosuppressed individuals.

We thank the Reviewer for the support.

Major points:

1)The clearance of SARS-CoV-2 is associated with an increase in FRNT titer, a decrease in HIV genome copies, and an increase in CD4 count. The authors have highlighted that clearance was associated with an increase in neutralizing activity. However, it might be possible that CD4+ cell increase may lead to increased CD8+ activity. Is it possible to look at CD8+ T cell activity in these individuals?

This is a critical point and we thank the Reviewer for encouraging us to address it. We have now used spike peptide pools specific to the infecting variant to stimulate T cells and determine the frequencies of CD8 and CD4 SARS-CoV-2 specific cells by flow cytometry using staining for interferon-gamma. We present our gating strategy and the frequencies in control, non-immunosuppressed participants in a new supplementary Figure 6:

As can be seen in Figure S6B above, this approach (now described in the Materials and Methods) readily detects SARS-CoV-2 specific CD8 and CD4 T cells. A new Figure 3 presents the unstimulated control and the frequencies of CD4 and CD8 cells in the samples from control participants at two timepoints – after infection and after vaccination (we expected and mostly obtained a further rise in frequencies after vaccination):

This is now described in the Results, lines 253-264:

“To determine whether SARS-CoV-2 specific T cell responses were present, we used stimulation with spike peptide pools³⁶, where the pool used was based on the sequence of the infecting variant. Spike-specific responding T cells were detected using flow cytometry, measuring interferon-gamma (IFN- γ) production (Figure 3A, see Figure S6A for the gating strategy). First, we tested responses post-infection (pre-vaccination) and post-vaccination in five study participants without HIV or with controlled HIV, approximately matched for age and sex to the advanced HIV disease group (Table S4). We used a post-vaccination timepoint as we expected vaccination may increase SARS-CoV-2 specific T cell frequencies. In this group, all participants had SARS-CoV-2 specific CD4 T cells post-infection (timepoint designated T1 in Figure 3A-B), which increased after vaccination for 3 out of 5 participants (timepoint T2 in Figure 3B, Figure S6 shows flow cytometry results for each participant). CD8 responses were also present post-infection in all except one participant (Figure 3B, Figure S6B).”

We next proceeded to test the advanced HIV disease participants for SARS-CoV-2 specific responses. Not surprisingly, there were not enough cells to determine if there were CD4 specific responses. However, we did not find SARS-CoV-2 specific CD8 cells either pre- or post-SARS-CoV-2 clearance. This is now presented in Figure 3C:

This important conclusion is highlighted in the Abstract, lines 38-41:

“SARS-CoV-2 clearance was associated with the emergence of neutralizing antibodies but not SARS-CoV-2 specific CD8 T cells, while CD4 T cell responses could not be determined because of low cell numbers.”

2) It is not clear why neutralizing antibodies appear after 100-200 days of infection. One explanation is that an increase in CD4+ T cells might have improved B cell affinity maturation, leading to potent neutralizing antibody production. However, those individuals might still be responding to infection and generating non-neutralizing antibodies even when they have low levels of CD4+ T cells. It will be important to look at different antibody isotypes (IgM, IgG, and IgA) to see if those individuals were making antibodies to SARS-CoV-2.

We have now checked IgA and IgM isotypes and our interpretation of the results is that IgA does not seem to associate with clearance. The results are presented in a new supplementary Figure 5:

3)The clearance of SARS-CoV-2 and HIV RNA coincides with the initiation of ART. Is there any direct effect of ART on SARS-CoV-2? Has it been shown that ART does not inhibit SARS-CoV-2 replication? If not, an in vitro test can be performed to show the effect of these drugs on SARS-CoV-2.

This is an important point which we have not sufficiently considered in the initial version of the manuscript. Our feeling was that inhibition of SARS-CoV-2 by antiretroviral therapy was disproven, but there is evidence either way based on epidemiological studies (now discussed on lines 202-205 of the Results section). We therefore tested it in vitro by taking one of the co-formulated TLD (tenofovir/ lamivudine/ dolutegravir) pills used as first line ART in the study, crushing and dissolving it, then testing the TLD solution for inhibition in both HIV and SARS-CoV-2 cellular infections. We found that TLD was a powerful inhibitor of HIV infection in a concentration dependent manner but had no detectable effect on SARS-CoV-2 infection.

This is now presented in a new supplementary Figure 3:

Minor points

1)Line 50 – the sentence seems incomplete

This sentence was removed.

2)Line 66-67 – the sentence needs to be rephrased

The sentence has been rephrased as (lines 85-88):

“There can be multiple reasons for immunosuppression^{7,16,42,48,49}. One cause of immunosuppression which has been shown in case studies to lead to SARS-CoV-2 long-term infection and evolution is uncontrolled HIV infection resulting in extensive CD4 T cell depletion, termed advanced HIV disease^{4-6,15}.”

3) Show correlation plots to show the correlation between SARS-CoV-2 genome copies, neutralizing antibodies, CD4+ T cell count, and HIV genome copies.

There are not enough autologous virus neutralization data points to make an informative correlation plot unless we combine participants, and this would mask the heterogeneity of participant responses. We therefore prefer to keep the SARS-CoV-2 titer/neutralizing antibody association figure the way it is.

4) Fig 2. X-axis label missing

It's at the bottom of the Fig 2 in our version but easily missed. We have therefore added it to each row.

5) How many times FRNT were performed?

We have now added this to the legend of Figures 2,4,5,6:
Figure 2 legend:

“Numbers above bars are geometric mean FRNT₅₀, and error bars are geometric mean standard deviations of FRNT₅₀ determined from 2-4 independent experiments.”

Figure 4 legend:

“All FRNT₅₀ values are geometric means from 2-3 independent experiments.”

Figure 5 legend:

“Numbers above the points and horizontal bars are geometric means for the group with FRNT₅₀ values determined once for each animal.”

Figure 6 legend:

“The numbers above the points and horizontal bars are geometric means for the group with FRNT₅₀ values determined once for each participant/virus combination.”

6) Can you include the conclusion for every result section? The way it is written now makes it very hard to follow the paper. If you include a conclusion for every section, it would be easier for the readers to transition from one section to another.

We thank the Reviewer for the suggestion. We now include the following summaries at the end of each section of the Results:

Section: Advanced HIV disease leads to long-term SARS-CoV-2 infection and evolution

Summary paragraph: “We conclude that these individuals with advanced HIV disease have a significantly prolonged SARS-CoV-2 infection. Secondly, during the time of prolonged SARS-CoV-2 infection, there is an accumulation of mutations, some of which are known to lead to escape from neutralizing antibodies.”

Section: Clearance of prolonged SARS-CoV-2 infection associates with emergence of neutralization

Summary paragraph: “In summary, we found an association between SARS-CoV-2 clearance and emergence of a neutralizing response. Complete HIV suppression was not required, and neutralization was likely mediated by IgG but not IgA isotypes, with the limitation that the antibody isotypes were measured in the blood and not at the infection site.”

Section: No association with SARS-CoV-2 specific CD8 T cell responses

Summary paragraph: “In summary, we did not find evidence that SARS-CoV-2 specific CD8 T cell responses are involved in SARS-CoV-2 clearance during recovery from advanced HIV mediated immunosuppression.”

Section: Poor vaccine elicited neutralization in advanced HIV disease participants with HIV viremia

Summary paragraph: “Therefore, while mRNA vaccination was effective in eliciting a neutralization response in participants without advanced HIV disease as well as those with advanced HIV disease but who already controlled their HIV infection, it did not perform well in eliciting SARS-CoV-2 neutralization in the two advanced HIV disease participants with HIV viremia.”

Section: Hamster infection shows evolved virus from Delta variant infection is antigenically distinct

Summary paragraph: “Our conclusion from this data is that prolonged SARS-CoV-2 infection in advanced HIV disease immunosuppression can lead to the evolution of SARS-CoV-2 which has extensive antigenic differences relative to both past and currently circulating strains.”

Section: Virus evolved from Delta escapes Delta but not Omicron XBB-elicited neutralization

Summary paragraph: “In summary, our results show that while the most evolved virus sampled in this study was antigenically distinct from the Omicron XBB.1.5 subvariant, this virus did not escape neutralization resulting from XBB-derived subvariant infections or relatively recent hybrid immunity.”

Reviewer #3:

Data are clearly presented. Conclusions drawn are well-supported. Findings of substantial evolution of SARS-COV-2 virus in immunosuppressed subjects over long infection period is consistent with other researchers finding. The strong association between neutralizing response and SARS-COV-2 virus clearance is important for the field. The observations that a “normal” vaccine responses can be elicited in people living with advanced HIV disease provided that viremia is controlled and CD4 T cell population is reconstituted should be informational for clinical practices involving advance HIV infections.

We thank the Reviewer for the support.

Questions:

1. The association between timing of neutralizing antibody response and virus clearance if clear in the study. However, timing of neutralizing antibody response also coincide with CD4 T cell reconstitution in most cases. Have SARS-COV-2 specific T cell responses being studied in these participants? HIV viremia control can be associated with both better CD4 health and CD8 T cell response. It would be great if we can see whether T cell response is also involved

in the SARS-COV-2 clearance. I do understand such investigation can be limited by specimen availability though.

The Reviewer raises a very important point. While the limited PBMC samples allowed us to do only one run and CD4 cell numbers were too low to detect possibly low frequencies of SARS-CoV-2 specific CD4 T cells, we were able to look at sufficient CD8 T cells to determine that all advanced HIV disease participants did not have detectable SARS-CoV-2 specific CD8 T responses at clearance. Details of the experiments are described below, and are included as a new main figure, a new supplementary figure, and new sections in the manuscript.

We summarize the finding in the Abstract, lines 38-41:

“SARS-CoV-2 clearance was associated with the emergence of neutralizing antibodies but not SARS-CoV-2 specific CD8 T cells, while CD4 T cell responses could not be determined because of low cell numbers.”

The methodology is described in a new section in the Materials and Methods (lines 649-666). We show the gating strategy in a new supplementary Figure 6A:

We present the results from the unstimulated and peptide stimulated cells from a control (non-advanced HIV disease) participant in a new Figure 3A. This shows that the methodology detects SARS-CoV-2 specific cells:

We then use control participants who were SARS-CoV-2 infected and then vaccinated to investigate the SARS-CoV-2 specific CD4 and CD8 T cell responses in the absence of advanced HIV disease mediated immunosuppression.

The raw results are presented in supplementary Figure 6B and show clear SARS-CoV-2 specific CD4 responses in all participants. Most are stronger post-vaccination, as expected. We also see well clear SARS-Cov-2 specific CD8 T cell responses in 4 out of 5 control participants:

The results for the control participants are graphed in Figure 3B:

In contrast to the control participants, we detect no SARS-CoV-2 specific CD8 responses before or after SARS-CoV-2 clearance in the advanced HIV group. Results are shown in Figure 3C:

These results are described in a new section in the Results on lines 252-281:

“No association with SARS-CoV-2 specific CD8 T cell responses

To determine whether SARS-CoV-2 specific T cell responses were present, we used stimulation with spike peptide pools³⁶, where the pool used was based on the sequence of the infecting variant. Spike-specific responding T cells were detected using flow cytometry, measuring interferon-gamma (IFN- γ) production (Figure 3A, see Figure S6A for the gating strategy). First, we tested responses post-infection (pre-vaccination) and post-vaccination in five study participants without HIV or with controlled HIV, approximately matched for age and sex to the advanced HIV disease group (Table S4). We used a post-vaccination timepoint as we expected vaccination may increase SARS-CoV-2 specific T cell frequencies. In this group, all participants had SARS-CoV-2 specific CD4 T cells post-infection (timepoint designated T1 in Figure 3A-B), which increased after vaccination for 3 out of 5 participants (timepoint T2 in Figure 3B, Figure S6 shows flow cytometry results for each participant). CD8 responses were also present post-infection in all except one participant (Figure 3B, Figure S6B).

We next tested for the presence of SARS-CoV-2 specific T cells in the advanced HIV disease participants. Because some of the samples had insufficient cells, we could not choose the same timepoints used to measure neutralizing antibodies. Nevertheless, we were able to test cells from two timepoints, one before and close to SARS-CoV-2 clearance (T1), and one post-clearance, T2 (Table S5). As expected, at T1 we observed that the proportion of CD4 T cells was low (ranging from 0.9 to 8.8% of total CD3+ T cells, Figure 3C). Consistent with ART-associated T cell reconstitution, at T2 (after ART initiation and in most cases HIV suppression), CD4 frequencies increased in all samples except for participant 27. In contrast, CD8 T cells were preserved at both timepoints, with high cell numbers.

We did not observe SARS-CoV-2 specific CD8 T cells in any of the participants with advanced HIV (Figure 3C). SARS-CoV-2 specific CD4 T cells were undetectable in all but one sample (participant 209, T2 sample), where they were present at a low frequency (0.017%). However, we acknowledged that the absence of detectable SARS-CoV-2 specific CD4 T cells in patients with severe lymphopenia may be attributed to the limited number of CD4+ T cells available for flow cytometry analysis. In summary, we did not find evidence that SARS-CoV-2 specific CD8 T cell responses are involved in SARS-CoV-2 clearance during recovery from advanced HIV mediated immunosuppression.”

2. Neutralizing antibody response development is associated with CD4 T cell reconstitution in most cases. Therefore CD4 T cell count also strongly associated with virus clearance in most

cases. However in Pt 255, CD4 T cells returned to normal range at D237, whereas neutralizing response and virus clearance didn't happen until D293. It may worth noting in discussion that at least in this case, CD4 T cell reconstitution itself isn't sufficient for virus clearance.

This is an important point, and we now add these details to lines 245-247 of the Results:

"...although participant 255, who did not have a suppressed HIV viral load at clearance, showed CD4 T cell reconstitution which plateaued at day 237 post-diagnosis, a time when SARS-CoV-2 was not yet cleared. Clearance was detected on day 293 (Figure 2D, second from right)."

We add this point to the Discussion on lines 405-411:

"Emergence of the neutralizing antibody response was associated with CD4 T cell reconstitution, as measured by CD4 T cell concentrations in the blood, in 4 out of 5 advanced HIV disease participants. However, in participant 255, CD4 T cell reconstitution peaked at day 237 post-SARS-CoV-2 diagnosis, whereas the neutralizing response and virus clearance happened on day 293. At least for this case, CD4 T cell reconstitution was not sufficient for virus clearance, or clearance was delayed relative to CD4 reconstitution."

Minor comments:

3. Line 193 "Participant 255 cleared SARS-CoV-2 at the vaccine baseline visit..." and then Line 195 "Participant 255 still had SARS-CoV-2 infection at vaccination". This is very confusing. Is the Participant 255 on line 195 supposed to be Participant 209?

We thank the Reviewer for picking this up. Corrected on lines 299-300 in the current version:

"In participants 255 and 209, HIV viremia was present at vaccination (Figure 1C). Participant 255 cleared SARS-CoV-2 at the vaccine baseline visit, before vaccination was administered (Figure 1B). Participant 209 had SARS-CoV-2 infection at vaccination (second dose, Figure 1B)."

4. Line 198 "but not at two weeks post-vaccination, the first post-vaccination timepoint tested and expected peak...". Is this referring to only after the first vaccination? It looks like neutralizing response did increase at the first time point tested after the 2nd vaccination.

We thank the Reviewer for picking this up. The neutralizing antibody response did increase for BA.1 but not the other viruses. This is corrected on lines 303-307 in the current version:

"For 209, neutralization capacity remained below the level of quantification for all viral strains except Omicron BA.1 at two weeks post-second dose. BA.1 neutralization did increase slightly ($FRNT_{50}=37$ to $FRNT_{50}=82$) two weeks post-vaccination which is the expected peak of the vaccine elicited neutralization response (Figure 4A, right panel)."

5. Line 211 "and this was also observed for participant 209". This refers to "strong increase in anti-spike antibodies but not neutralization". However for Pt 209, there is not really a strong increase in anti-spike antibodies following vaccination. The peak binding response that shows an increase is at Day 0 of 1st vaccination, which is most likely from infection. The binding response did not increase until much later on.

We thank the Reviewer for pointing this out. Indeed, there was no increase of anti-spike antibodies post-vaccination for participant 209. We have now corrected this description (lines 313-315 in the current version):

“Anti-spike binding antibody levels mostly mirrored the neutralizing antibody response to the different viral strains in these participants as well as in 4 out of 5 advanced HIV participants. The exception was 255, one of two participants with HIV viremia at vaccination.”

Sincerely yours,

Alex Sigal, Africa Health Research Institute

REVIEWERS' COMMENTS

Reviewer #1 (Remarks to the Author):

OVERALL

The manuscript is much improved. The addition of the T cell work with the finding that CD8 responses are not associated with clearance and the incidence of chronic infection in advanced HIV are both very important observations.

INTERPRETATION

Line 438 "Therefore, SARS-CoV-2 vaccination in the absence of HIV suppression may not be a viable strategy to prevent prolonged infections." – suggest to reword this to a more active statement " These data support the principle of starting ARVs in parallel with or prior to SARS-CoV-2 vaccination in patients with HIV".

Suggest to add a sentence to emphasise the importance of this study re VOCs - if there is a 54% risk of prolonged infection of SARS-CoV-2 in people with advanced HIV, this means that there is likely to be a long-term public health risk associated with the generation of escape mutants/VOCs in this group.

MINOR SUGGESTION

Line 65: "It is much more difficult for viruses to escape T cell recognition using mutations." – please clarify – I understand what the authors are trying to say – but the sentence is ambiguous, please reword.

Reviewer #2 (Remarks to the Author):

The authors have addressed the concerns raised by this reviewer. They have performed additional experiments to demonstrate the role of CD4 and CD8 T cell response in viral clearance. They also performed experiments to look at the antibody isotypes. They performed an in vitro assay to show that ART drugs do not inhibit SARS-CoV2 replication which further validates the conclusions made in the paper. Overall, these new experiments have improved the paper significantly and this paper can be accepted for publication.

Reviewer #3 (Remarks to the Author):

The authors addressed questions I raised adequately. The new experimental data on T cell response and effects on antivirals on SARS-CoV-2 replication are very informational. The analysis on the frequencies of long-term COVID in PLWH is also valuable. I have no further questions or concerns.

Reviewers 2 and 3 had no further comments and supported publication. The comments of Reviewer 1 are addressed below:

1) Line 438 “Therefore, SARS-CoV-2 vaccination in the absence of HIV suppression may not be a viable strategy to prevent prolonged infections.” – suggest to reword this to a more active statement “ These data support the principle of starting ARVs in parallel with or prior to SARS-CoV-2 vaccination in patients with HIV”.

Added the statement as suggested on lines 436-438: “These data therefore support the principle of starting ARVs in parallel with or prior to SARS-CoV-2 vaccination in patients with HIV.”

2) Suggest to add a sentence to emphasise the importance of this study re VOCs - if there is a 54% risk of prolonged infection of SARS-CoV-2 in people with advanced HIV, this means that there is likely to be a long-term public health risk associated with the generation of escape mutants/VOCs in this group.

Added to the conclusion in the abstract: “...and that suppressive ART is necessary to curtail evolution of co-infecting pathogens to reduce individual health consequences as well as public health risk linked with generation of escape mutants.”

3) Line 65: “It is much more difficult for viruses to escape T cell recognition using mutations.” – please clarify – I understand what the authors are trying to say – but the sentence is ambiguous, please reword.

This has been changed to (line 64): “However, several factors constrain the selection of viral escape mutations from T cell mediated immunity.”